# Chalcones and Gastrointestinal Cancers: Experimental Evidence

**DOI:** 10.3390/ijms24065964

**Published:** 2023-03-22

**Authors:** Radka Michalkova, Martin Kello, Martina Cizmarikova, Annamaria Bardelcikova, Ladislav Mirossay, Jan Mojzis

**Affiliations:** Department of Pharmacology, Faculty of Medicine, Pavol Jozef Šafárik University, 040 01 Košice, Slovakia

**Keywords:** chalcones, colorectal cancer, gastric cancer, cell death, angiogenesis, signaling pathway, inflammation, *Helicobacter pylori*, multidrug resistance, reactive oxygen species

## Abstract

Colorectal (CRC) and gastric cancers (GC) are the most common digestive tract cancers with a high incidence rate worldwide. The current treatment including surgery, chemotherapy or radiotherapy has several limitations such as drug toxicity, cancer recurrence or drug resistance and thus it is a great challenge to discover an effective and safe therapy for CRC and GC. In the last decade, numerous phytochemicals and their synthetic analogs have attracted attention due to their anticancer effect and low organ toxicity. Chalcones, plant-derived polyphenols, received marked attention due to their biological activities as well as for relatively easy structural manipulation and synthesis of new chalcone derivatives. In this study, we discuss the mechanisms by which chalcones in both in vitro and in vivo conditions suppress cancer cell proliferation or cancer formation.

## 1. Introduction

Digestive tract cancers are major causes of morbidity and mortality from cancer across the world today. Colorectal cancer (CRC) and gastric cancer (GC) are the most common digestive tract cancers and represent the second and fourth leading causes of cancer-related deaths, as reported by GLOBOCAN 2020 [1].

Chemotherapy is often used to treat both CRC and GC, but it has numerous disadvantages, including toxicity, chemoresistance induction and the need for supportive care [2,3]. Therefore, it is extremely important to identify novel compounds (either natural or synthetic) with anticancer effects and favorable safety levels to combat cancers, including CRC and GC.

Nature has been, is and always will be an important source of compounds with various biological activities, including anticancer activity [4]. It is worth noting that some conventional anticancer drugs, including taxanes, vinca alkaloids and epipodophyllotoxins, also originate from nature [5,6].

A large family of natural compounds are polyphenols. These plant secondary metabolites are found in vegetables, fruits and seeds and are broadly distributed throughout plants [7]. They control several plant functions, such as protection from pathogens, UV irradiation and oxidative stress [8]. In addition to plant protection, numerous studies have documented the beneficial effects of these compounds on human health, including anticancer, anti-inflammatory, antioxidant and cardioprotective activities [9,10].

In recent decades, a large amount of attention has been focused on chalcones, which are the metabolic precursors of flavonoids. Not only are they abundant in nature and relatively easy to synthesize but their structural simplicity also carries a great deal of potential for the synthesis of a plethora of derivatives with pleiotropic bioactivities due to interactions between numerous macromolecules and various signaling pathways [11,12]. Many studies have described the broad spectrum of biological activities of chalcones, including anti-inflammatory, antimicrobial, antiparasitic, antidiabetic and antioxidant activities [13,14,15,16,17]. Moreover, the anticancer effects of both natural and synthetic chalcones have also been intensively studied [18,19,20].

Growing data have indicated that chalcones are multitarget anticancer compounds. The possible molecular mechanisms involved in the tumor-suppressive activity of these compounds are shown in Figure 1.

In this paper, we summarize the cellular and molecular mechanisms of natural and synthetic chalcones that are related to their antiproliferative and anticancer effects in colorectal cancer and gastric cancer.

## 2. Molecular and Cellular Mechanisms of Action

### 2.1. Effects on Cell Cycle and Apoptosis

Cell cycle arrest prevents the proliferation of damaged cells. During this process, cells facilitate DNA repair or, if the DNA defects are too extensive, other signaling pathways are activated for cell removal [21]. Several chalcones have been found to block cancer cell proliferation via cell cycle arrest and the activation of cell death machinery. Despite their ability to inhibit the cell cycle in different phases [22,23,24], most chalcones have been observed to block cell cycle progression at the G2/M phase.

At our laboratory, we evaluated the mechanisms of numerous synthetic chalcone derivatives using either Caco-2 or HCT116 human colorectal cell lines. Flow cytometric analysis showed that several chalcones (e.g., (E)-2-(4′-methoxybenzylidene)-1-benzosuberone, (E)-2-(2′,4′-dimethoxybenzylidene)-1-tetralone and (2 E)-3-(acridin-9-yl)-1-(2,6-dimethoxyphenyl) prop-2-en-1-one) caused an accumulation of cancer cells in the G2/M phase, with a concomitant increase in cells with sub-G0/G1 DNA content (a marker for apoptosis) [25,26,27]. Three decades ago, chalcones were discovered to be antimitotic compounds [28]. Because the accumulation of cells in the G2/M phase could be a consequence of mitotic spindle dysregulation, we evaluated the effects of the studied chalcones on tubulin expression. Our results showed that the expression of tubulins was deregulated on both the genomic and protein levels.

The ability of chalcones to initiate cell cycle arrest at the G2/M phase has also been documented in numerous other publications. Dias et al. [29] synthesized several halogenated chalcones and their cell cycle analysis showed that the antiproliferative effect of the most potent compounds in HCT116 cells was associated with G2/M arrest and an increase in cells with sub-G0/G1 DNA content. Later, G2/M arrest and deregulated microtubular networks were also found in methoxychalcone-treated SW620 colon cancer cells [30]. These chalcones also modulated the levels of cell cycle regulatory proteins, as evidenced by the downregulation of cyclin D1 and cyclin A and the upregulation of cyclin B1. The G2/M arrest associated with the dysregulation of cyclin A and B1 expression has also been observed in xanthohumol (Xn)-treated HT-29 cells [31]. Recently, Moura et al. [32] documented the sulfonamide chalcone-induced G2/M arrest of colorectal adenocarcinoma metastatic cells with subsequent cell death. A detailed review of chalcones as tubulin polymerization inhibitors has also been released recently [33].

Mitotic spindle dysregulation and G2/M arrest often result in cell death. One of the hallmarks of cancer cells is defective apoptosis [34]. It is well known that the induction of apoptosis is one of the crucial mechanisms that inhibits cancer cell growth. This cell suicide process involves extrinsic and intrinsic (mitochondrial) pathways. The intrinsic pathway is associated with the modulation of Bcl-2 family protein activity, as well as the release of cytochrome *c* and other pro-apoptotic factors, with the subsequent activation of several caspases. On the other hand, in extrinsic pathway, activation of surface receptors leads to the activation of death signaling via the activation of initiator procaspases-8 and -10 [35].

Chalcones play important role in the induction of both intrinsic and extrinsic apoptosis pathways [36]. Several studies have demonstrated the ability of chalcones to induce the intrinsic apoptosis (mitochondrial) pathway. The structure and function of mitochondria are significantly changed in cells that have been exposed to apoptotic stimuli [37]. The modulation of Bcl-2 protein family activity leads to the permeabilization of mitochondrial outer membrane, the loss of mitochondrial membrane potential (MMP) and the subsequent release of proapoptotic factors, followed by caspase activation and cell death [38,39]. In various colon cancer cells, chalcone treatment leads to either the activation of the proapoptotic members of the Bcl-2 protein family (e.g., Bax and Bad) or the suppression of antiapoptotic proteins (e.g., Bcl-xL and Bcl-2), resulting in the release of several proapoptotic proteins (e.g., cytochrome and Smac/DIABLO) and the activation of initiator caspase 9 or execution caspases 3/7. Moreover, a decrease in MMP has also often been detected [26,27,31,40,41].

Another important player in apoptosis regulation is the protein p53, which is a tumor suppressor activated by DNA damage. Its phosphorylation (i.e., activation) and subsequent caspase activation have been observed in sappanchalcone-treated HCT116 cells. On the other hand, colon cancer cells with mutated p53 have been found to be insensitive to sappanchalcone treatment [42]. A study by Shin et al. [43] showed a high level of p53 and apoptosis induction in chalcone-treated HCT116 cells. Interestingly, they also found apoptosis induction in p53-null HCT116 cells, suggesting that both p53-dependent and p53-independent mechanisms can play roles in apoptosis that is induced by this chalcone. In addition, several other publications have revealed an association between p53 activation and apoptosis induction in chalcone-treated colon cancer cells [44,45,46,47]. Furthermore, both natural and synthetic chalcones have been studied as potential anticancer agents in gastric carcinogenesis models [18,48].

Similar to their activity in colon cancer cells, chalcones also mostly stop cell cycle progression at the G2/M phase in gastric cancer cells. Guan et al. [49] documented a strong antiproliferative effect of the newly synthesized quinoline chalcone derivative 12e against the MGC-803 gastric cancer cell line. Among other effects, this chalcone-induced G2/M arrest was followed by the activation of caspases-3 and -9, indicating the mitochondrial pathway of apoptosis. The cumulation of cells in the G2/M phase of the cell cycle has also been observed in 2′,4′-dihydroxychalcone-treated MGC-803 cells. This effect has been associated with the down-regulation of survivin and the activation of caspase-3 [50]. The antiproliferative effect of licochalcone A (LiA; a natural chalcone isolated from the root of Chines licorice, *Glycyrrhiza inflata*) in different gastric cancer cell lines has also been associated with the inhibition of the cell cycle at the G2/M phase with the simultaneous down-regulation of cyclin A and cyclin B expression [51].

Additionally, it has been reported that the antiproliferative effects of other chalcones, such as isoliquiritigenin analog [52] and flavokawain B [53], are associated with G2/M cell cycle arrest. Furthermore, it is important to mention that several chalcones also initiate cell death via the activation of the extrinsic apoptosis pathway [54,55,56,57] or the induction of non-apoptotic cell death [53,58,59,60]. General pathways involved in the cell cycle and apoptosis are presented in Figure 2.

### 2.2. Modulation of Signaling Pathways

Several signaling pathways that are involved in cell survival and death frequently mutate during cancer progression [61].

#### 2.2.1. Wnt/β-Catenin Signaling Pathway

The Wnt/β-catenin signaling pathway, also known as the canonical Wnt signaling pathway, plays a crucial role in numerous biological processes, such as embryonic development, the regulation of the cell cycle, apoptosis and many others. On the other hand, the dysregulation of the Wnt/β-catenin pathway is often associated with a variety of diseases, including cancer [62,63]. In colorectal cancers, the Wnt/β-catenin pathway is one of the most important signaling pathways [64]; therefore, the Wnt/β-catenin signaling pathway is an attractive target for cancer treatment and several small inhibitors targeting the Wnt/β-catenin pathway have been examined in preclinical and clinical studies [65,66,67]. In addition, some studies have indicated that chalcones are also capable of acting on this signaling pathway. Brousochalcone A, a prenylated chalcone isolated from paper mulberry, decreases β-catenin levels by promoting its proteasomal degradation [68]. A similar mechanism of action has also been reported in cardamonin. This chalcone down-regulates β-catenin levels via proteasome-mediated degradation in SW480 cells [69]. Another chalcone, lonchocarpin, has also been reported to suppress colorectal cancer cell proliferation in vitro. This effect has been associated with the inhibition of the Wnt signaling pathway via the blockade of β-catenin nuclear translocation, with the subsequent inhibition of its interaction with DNA. Moreover, lonchocarpin has also been shown to inhibit β-catenin signaling in vivo in a *Xenopus laevis* embryo model [70]. Other negative modulators of the Wnt/β-catenin pathway are derricin and dericcidin, which are chalcones isolated from *Lonchocarpus sericeus.* Both of these chalcones have been found to inhibit Wnt signaling in colorectal HCT116 cells, with subsequent cell viability reduction and cell cycle arrest at the G2/M phase [71]. Later, Li et al. [72] documented the antiproliferative effect of isobavachalcone (IBC) (a natural chalcone isolated from *Psoralea corylifolia*, Buguchi) in HCT116 and SW480 colon cancer cells. This effect was associated with the induction of the intrinsic pathway of apoptosis. Their Western blot analyses showed the down-regulation of the Wnt/β-catenin pathway. Further analyses showed that IBC also inhibited the phosphorylation of GSK-3β and AKT, which are upstream proteins in the β-catenin signaling.

Cardamonin has also been reported to induce apoptosis via the inhibition of the Wnt/β-catenin signaling pathway in BGC-823 and BGC-823 5-fluorouracil (5-FU)-resistant gastric cancer cells [73]. In addition, in combination with 5-FU, cardamonin increases the sensitivity of resistant cells to 5-FU and down-regulates several Wnt target genes, including β-catenin, TCF4 (transcription factor 4) and cyclin D1.

#### 2.2.2. Nuclear Factor Kappa B Signaling Pathway

The nuclear factor kappa B (NF-kB) signaling pathway plays a key role in the regulation of numerous genes and proteins involved in broad spectrum biological activities, such as inflammation, immune response and cell death and survival. The abnormal activation of the NF-kB signaling has been reported in different tumor types [74].

Licochalcone A has been reported to suppress the proliferation of colon cancer cells. Further Western blot and Rt-PCR analyzes have shown that LiA negatively modulates NF-kB signaling and that this effect is closely related to the inhibition of p65 phosphorylation. It is important to mention that p65 phosphorylation is crucial for the transcriptional activity of NF-κB [75]. Recently, Papierska et al. [76] documented the potential of newly synthesized thioderivative chalcones to modulate NF-κB activity. This effect was associated with a decrease in the p65 nuclear fraction in chalcone-treated HCT116 cells. Moreover, the expression of the gene encoding p65 also significantly decreased in treated cells. The anticancer effect associated with the inhibition of NF-κB signaling and p65 translocation has also been observed in vivo in cardamonin-treated mice [77]. Cardamonin has also been reported to reduce the resistance of colon cancer cells to 5-FU. In HCT116 5-FU-resistant cells, cardamonin has been found to induce apoptosis and suppress proliferation. This activity has been associated with the suppression of testes-specific protease 50 (TSP50) and NF-κB protein expression. Because TSP50-induced cell proliferation and tumorigenesis are dependent on the activation of the NF-kB signaling pathway [78], it has been suggested that the antiproliferative effect of cardamonin could be related to the inhibition of TSP50/NF-κB protein expression [79]. Furthermore, the ability to suppress cancer proliferation via the inhibition of NF-κB signaling has also been reported in several other chalcones, such as 4-Boc-piperidone chalcones [80], bichalcone analogs [81], IBC [82], hydroxysafflor yellow A [83] and butein [84].

#### 2.2.3. Mitogen-Activated Protein Kinase (MAPK) Signaling Pathway

The mitogen-activated protein kinase (MAPK) pathway is involved in the regulation of different biological processes, such as gene expression and cell proliferation and survival. In cancer, the aberrant activation of MAPK signaling is often associated with the uncontrolled proliferation of cells and defective apoptosis [85]. There are several components in MAPK signaling, although extracellular regulated kinases (ERK1/2), c-Jun N-terminal kinase (JNK) and p38 have been studied the most.

Cardamonin has also been reported to be an inhibitor of β-catenin and NF-κB signaling. Another mechanism of its antiproliferative effect is related to its ability to activate JNK via the stimulation of p53. As many authors have suggested, the activation of the p53/JNK pathway is essential for cardamonin-induced autophagy and the suppression of HCT116 colon cancer cells [86]. Furthermore, the antiproliferative effect of isoliquiritigenin (ISL) has also been associated with the increased phosphorylation of JNK and ERK as a result of the increased production of reactive oxygen species (ROS) [87]. In addition, ERK1/2 activation has also been observed in flavokawain C-treated HCT116 cancer cells [88]. A similar effect on ERK1/2 activity, i.e., increased phosphorylation, has also been documented in HCT116 cells exposed to a synthetic acridine–chalcone hybrid, along with the increased activity of other members of the MAPK signaling pathway (p38 and JNK) [27]. On the other hand, LiA has been reported to be an inhibitor of JNK, specifically JNK1 [89]. This chalcone inhibits colon cancer cell proliferation and colony formation. Moreover, in vivo experiments have shown that LiA suppresses the growth of HCT116 xenografts due to decreased JNK phosphorylation in tumor tissues. The suppression of the ERK signaling pathway and the associated viability decrease have also been reported in Xn-treated HT-29 colon cancer cells. These effects have been associated with the inhibition of Ras and MEK, as well as the other proteins in the MAPK signaling pathway [31]. In addition, a decrease in ERK1/2 protein expression has also been found in 3-deoxysappanchalcone-treated colon cancer cells, along with the simultaneously increased expression of pro-apoptotic proteins [90].

The decreased phosphorylation of Akt and ERK proteins has also been observed in IBC-treated MGC803 gastric cancer cells. As documented by Jin and Shi [91], IBC-induced apoptosis in cancer cells was associated with the upregulation of Bax and the down-regulation of Bcl-2 proteins. As has been suggested, the inhibition of these signaling pathways is involved in the IBC-induced apoptosis of gastric cancer cells. Furthermore, the antimetastatic effect of chalcones could be related to the inhibition of the phosphorylation of JNK1/2 and focal adhesion kinase (FAK). Lin and Shin [92] also reported the decreased adhesion, migration and invasion of AGS gastric cancer cells exposed to this chalcone. The inhibition of JNK and FAK phosphorylation has been associated with the down-regulation of matrix metalloproteinase-2 and -9 expression, which are enzymes involved in extracellular matrix degradation.

Additionally, other signaling pathways have been documented as targets for chalcones, including PI3K/Akt/mTOR [52,88,93,94,95], COX-2 [96,97], STAT-3 [76,98,99,100] and Nrf2 [76].

The effect of chalcones on selected signaling pathways are shown in Figure 3.

### 2.3. Enzyme Inhibition

Topoisomerases play essential roles in DNA replication by preventing deleterious excessive supercoiling. Topoisomerase inhibitors are commonly used in anticancer therapy, including irinotecan, topotecan (topoisomerase I inhibitors; TOPO I), etoposide and teniposide (topoisomerase II inhibitors; TOPO II) [101].

Recently, Mohammed et al. [102] described the antiproliferative effects of novel urea-ciprofloxacin–chalcone hybrids on HCT116 colon cancer cells. These effects were associated with G2/M cell cycle arrest and apoptosis induction. Further analyses showed that the most potent hybrids inhibited TOPO I and TOPO II, which was comparable to clinically used drugs. Later, they also reported significant inhibitory activity against TOPO I and TOPO II in triazole linked ciprofloxacin chalcone-treated colon cancer cells. This anti-topoisomerase effect has also been associated with DNA damage, G2/M arrest and tubulin dysregulation [103]. Furthermore, several natural and synthetic chalcones have also been reported to be inhibitors of TOPO I [104,105] or TOPO II [106,107].

Histone deacetylase (HDAC) plays an important role in the epigenetic control of gene transcription and the consequences of its inhibition include cancer cell cycle arrest, cell death, the suppression of neovascularization and the modulation of immune response [108]. Furthermore, high levels of HDAC have been observed in both colorectal cancer and gastric cancer [109,110]. Today, several HDAC inhibitors have either been approved for cancer treatment or have reached different phases of clinical evaluation [111,112]. Several lines of evidence have indicated that both natural and synthetic chalcones possess the ability to inhibit HDAC and that some of them are even more effective than the inhibitors that are currently used in clinical practice [113,114,115].

Additionally, several other enzymes have been identified as targets for chalcones, including aurora kinases [116,117], cyclooxygenases (COXs), 5-lipoxygenase [96,118], hexokinases II [119], matrix metalloproteinases [120,121], various tyrosin kinases [122,123] and NO synthase [124].

### 2.4. Antiangiogenic Effect of Chalcones

One of the key factors in cancer development and metastasis is angiogenesis [125]. The process of neovascularization is gently orchestrated by numerous positive and negative regulators of angiogenesis. Among them, the vascular endothelial growth factor (VEGF) and its receptor (VEGFR-2) are important targets in the treatment of various cancer types, including gastrointestinal tract cancers [126,127,128]. In addition, numerous studies have shown the potential of natural compounds, including chalcones, in blocking different steps during angiogenesis [129,130,131].

Because the antiangiogenic effect of chalcones was recently reported in a review by Mirossay et al. [132], we only mention findings from the last five years in the following section.

As aforementioned, VEGFR-2 is an important target for antiangiogenic drugs. Recently, Ahmed et al. [133] demonstrated the ability of novel chalcone–piperazine hybrids to block VEGFR-2. Among the tested chalcone derivatives, the inhibitory activity of the most active compound was comparable to that of sorafenib (an antiangiogenic drug that is used clinically). Moreover, this compound also arrested the cell cycle at the G2/M phase and induced apoptosis in HCT116 colon cancer cells.

Hypoxia-inducible factor (HIF)-1α plays an important role in cancer adaptation to hypoxic microenvironments [134]. Among other activities, HIF-1α also regulates angiogenesis via the up-regulation of several pro-angiogenic factors, such as VEGF, transforming growth factor-β, plasminogen activator-1 and erythropoietin. Due to its wide spectrum of regulatory activities, HIF-1α signaling is a promising target for anticancer therapy. In addition to synthetic compounds, several phytochemicals have also been reported to modulate HIF-1α activity [135]. Recently, Park et al. [136] revealed the ability of LiA to inhibit the hypoxia-induced activation of HIF-1α and the subsequent down-regulation of HIF-1α-related genes. The ability to suppress HIF-1α signaling has also been documented in both synthetic and natural chalcones [137,138,139].

Chorioallantoic membrane (CAM) assay is widely used to study the antiangiogenic potential of experimental compounds. Several chalcones have been reported to reduce blood vessels in CAM. The application of a synthetic chalcone (named CAB7β) has been shown to significantly suppress tubule formation, junctions and the length and size of vasculatures in comparison to untreated controls [140]. Later, González et al. [141] evaluated the antiproliferative and antiangiogenic effects of newly synthesized chalcone gold(I) conjugates. In a CAM assay, these chalcone derivatives have been found to disrupt the creation of new blood vessels.

Furthermore, human umbilical vein endothelial cells (HUVECs) have also often been used as models to study the effects of agents targeting angiogenesis. Tube formation and the migration and invasion of endothelial cells are crucial features of angiogenesis. Recently, Wang et al. [142] synthesized new chalcones based on 2-methoxyestradiol. Among other effects, they found that the most active compound was able to block the migration of HUVECs. This effect was associated with the reduced expression of the VEGF receptor, as well as the reduced phosphorylation of focal adhesion kinase and SHC, which are two proteins involved in endothelial cell migration. In addition, these compounds significantly suppressed angiogenesis in a CAM assay. Furthermore, newly synthesized α-substituted hetero-aromatic chalcone hybrids have been found to inhibit several steps of angiogenesis. In a wound healing model, chalcone 7m significantly inhibited HUVEC migration and also suppressed HUVEC invasion in a dose-dependent manner, as detected by a transwell assay. Additionally, HUVEC tube formation and length were dramatically inhibited in the chalcone-treated group. Furthermore, the in vitro results were then confirmed in vivo using a zebrafish embryo model [143]. Later, they also documented a similar antiangiogenic effect also for new α-fluorinated chalcone [144].

In vivo antiangiogenic effects have also been reported in zebrafish embryos and rabbit corneas exposed to ISL. In both models, ISL has been shown to reduce neovascularization. The addition of prostaglandin E_2_ partly reverses these effects, indicating that ISL inhibits angiogenesis, at least partly, via the inhibition of the COX pathway [145].

### 2.5. Chalcones and Inflammation

It is well known that tumors originate in areas that are infiltrated by cells from the immune system, i.e., areas that are repeatedly exposed to inflammation [146]. It is now believed that chronic inflammation is responsible for changing sensitive cells via neoplastic transformation. In general, the elderly are at the greatest risk of developing cancer as they have experienced multiple inflammatory responses induced by infectious agents and other stressors during their lifetimes. Long-term exposure to carcinogenic factors, including inflammation, is a cause of cancer.

The presence of immune cells in tumors demonstrates that cells with innate immunity actively model tumor processes. The processes of chronic inflammation (i.e., leukocyte infiltration and angiogenesis) are thought to play important roles in tumor progression. Tumor microenvironments (TMEs) modulate tumor growth and progression through the production of reactive oxygen species (ROS), epigenetic changes and the simultaneous promotion of tumorigenesis through the production of growth factors and pro-inflammatory cytokines [147]. Growing tumors influence TMEs via feedback pathways through the production of cytokines and chemokines. An increased risk of cancer has also been shown to be associated with chronic intestinal diseases, which can be linked to microbial infections or inappropriate diets.

As a result of the demonstrated link between inflammation and cancer, research has been devoted to the development of anti-inflammatory drugs that could play important roles in the treatment and prevention of cancer. Chronic and inflammatory conditions in the gastrointestinal tract, including ulcerative colitis (UC) and Crohn’s disease, are often associated with the development of CRC [148].

Many scientific studies have shown that chalcones and their flavonoid derivatives are effective anti-inflammatory substances that protect against the development of cancer. Their antimicrobial, antibacterial and antiparasitic activities have been proven in relation to CRC. In general, anti-cancer chalcones could simultaneously be considered as anti-inflammatory agents due to the fact that they increase the synthesis of anti-inflammatory cytokines [149,150].

The Toll-like receptor 4 (TLR4) is abundantly expressed in intestinal epithelial cells and is thought to play a key role in intestinal innate immunity [151]. It is also involved in the pathogenesis of inflammatory bowel disease, with TLR4 activation resulting in the nuclear translocation of transcriptional NF-κB and/or the activation of MAPKs, leading to the production of pro-inflammatory cytokines and chemokines [152]. Many studies have shown that NF-κB plays a major role in the development and progression of cancer because it regulates more than 400 genes that are involved in inflammation and carcinogenesis [153]. Therefore, it is desirable to down-regulate TLR4 expression, block the activation of the NF-κB and MAPK signaling pathways, and inhibit the pro-inflammatory gene expression of COX, prostaglandin E2, inducible nitric oxide synthase (iNOS) and numerous cytokines (e.g., IL-1β, IL-6, IL-8 and TNF-α).

Chalcones, such as LiA, are important inhibitors of TLR4-mediated NF-κB activation, with higher accumulation in Caco-2 cells [154]. Licochalcone A exudes anti-UC activity, in part by blocking the MAPK pathway [155]. Another natural chalcone, ISL, reduces the incidence of colitis that is associated with colorectal cancer through the modulation of gut microbiota and pro-inflammatory cytokines [156]. Furthermore, xanthoangelol D (25 mg/kg) reduces the disease activity index (DAI) of colitis and the dextran sulfate sodium-induced increases in colonic MCP-1, IL-1β and TNF-α levels [157]. The inhibition of NF-κB activity, p38-regulated/activated kinase (PRAK) and MAPK-activated protein kinase (MAPKAP-K)-3 suppression have been reported effects of flavokawain A, which is isolated from *Piper methysticum* [158].

The anti-inflammatory properties of chalcones due to the inhibition of NF-κB activity are related to the downregulation of iNOS, COX-2, TNF-α and IL-6. Scientific studies on chalcones have demonstrated their anti-inflammatory properties via the observation of multiple pro-inflammatory markers. Xn [159] and cardamonin [160] have been found to be significant COX-2 inhibitors that also simultaneously inhibit iNOS. Kim et al. demonstrated that ISL reduced iNOS, COX-2, TNF-α and IL-6 [161]. Similarly, LiA inhibited the expression of iNOS and COX-2 and suppressed the production of NO and PGE_2_ [162]. Chalcone 4,2′,5′-trihydroxy-4′-methoxychalcone is isolated from the heartwood of *D. odorifera* and has been found to inhibit the production of NO, the expression of COX-2 and nitric oxide synthase and the release of TNF-α and IL-1β [163]. Chalcones isolated from *Humulus lupulus*, such as xanthohumol B, xanthohumol D and dihydroxanthohumol, have shown the potential to inhibit LPS-induced NO production without any cytotoxicity, as well as inhibiting iNOS expression [164].

The knowledge of the structure and mechanisms of action of chalcones provides scope for their de novo synthesis. Many successful publications have documented evidence of the high efficiency of newly synthesized chalcones, analogs and derivatives. For example, Guazelli et al. [165] used a chalcone derivative (the flavonoid hesperidin methyl chalcone; HMC) to treat acetic-acid-induced colitis in a mouse model. The HMC treatment significantly reduced neutrophil infiltration, edema and colonic shortening. An increase in the anti-inflammatory state of the colon was also observed via the inhibition of the pro-inflammatory cytokines TNF-α, IL-6, IL-1β and IL-33, as well as the inhibition of NF-κB activation in the colon. Papierska et al. [76] demonstrated the significant effects of newly synthesized chalcone thioderivatives on the regulation of the expression of genes involved in inflammatory processes (e.g., NF-κB, STAT3 and NRF2) in colorectal carcinoma cells. The anti-cancer and anti-inflammatory effects of lonchocarpin [70] and quercetin [166] have been confirmed in colorectal cancer cells. IL-8 and MMP-7 [167] are also significantly involved in the inflammatory processes of intestinal tissues. Matrix metalloproteinase (MMP) inhibitors ameliorate dextran sulfate sodium-induced colitis [168]. The synthetic 2′,4′,6′-tris(methoxymethoxy) chalcone (TMMC) is an anti-inflammatory substance that has been shown to inhibit TNF-α-induced IL-8 and MMP-7 production in HT-29 human CRC cells [169]. Additionally, TMMC has been found to reduce NO production through the inhibition of NF-κB activation and iNOS expression in LPS-activated RAW 264.7 cells. Some chalcones, such as 2-methyl-4-phenylquinoline-chalcone analogs, are substances with broad-spectrum actions, not only in the sense of inflammation but also as antidepressants and analgesics [170]. Mahmoud et al. [171] demonstrated that novel polymethoxylated chalcones increased CRC cell apoptosis, inhibited angiogenesis and reduced the invasiveness and metastatic potential of CRC cells via downregulation of several epithelial-mesenchymal transition (EMT) markers including vimentin, fascin and β- catenin.

Inflammation is a polygenic process that is determined by several different genes. Thus, in order to control inflammation, it is important to influence the expression of several genes simultaneously. The commonly available inhibitors of pro-inflammatory gene expression are only effective for specific genes. Chalcones are interesting because of their broad spectrum of action, which affects the transcriptional levels of several pro-inflammatory molecules, including TNF-α, IL-6, IL-1β and IL-33, as well as the inhibition of NF-B, IL-8 and MMP activation.

### 2.6. Chalcone-Induced Oxidative Stress

Cellular physiological processes are inherently associated with the production of reactive oxygen species (ROS), which are involved in several signaling pathways [172]. When ROS formation is uncontrolled, oxidative stress occurs, which disrupts redox signaling and causes damage to biomolecules, at the expense of cellular antioxidant defense mechanisms [173]. In cells, oxidation–reduction homeostasis is ensured through the glutathione [174] and thioredoxin systems [175]. Moreover, several antioxidant enzymes are present in cells, including superoxide dismutase (SOD), catalase (CAT), glutathione peroxidase (GPx) and glutathione S-transferase (GST), which maintain redox status and protect the cells against oxidative and nitrosative stress. It is well known that several types of ROS are produced continuously by mitochondrial bioenergetics and oxidative metabolism, including superoxide anions (O_2_^−^), hydroxyl radicals (·HO), perhydroxyl radicals (·HOO), non-radical hydrogen peroxide (H_2_O_2_) and nitrogen (RNS) species, such as nitric oxide (·NO) and peroxynitrite (ONOO^−^) [176].

In cancer research and therapy, radiotherapy and chemotherapy are the possible approaches to the treatment of tumor cells, which kill tumor cells through the production of ROS. Additionally, the anti-cancer effects of several drugs have been connected with ROS generation and/or decreases in ROS attenuators and antagonists, as reviewed in [177]. As previously described, chalcones are promising antitumor substances of natural origin, which also possess pro-oxidant activity and pro-apoptotic effects that are associated with oxidative stress and the modulation of ROS-dependent signaling pathways and DNA damage [36]. It is well known that the antioxidant properties of chalcones are based on structural variations and diversity in their molecules. Stabilized radicals are formed during reactions between chalcones and ROS. Moreover, the increased residence time and diffusion distance of radicals that enable clearance via endogenous antioxidants, such as glutathione (GSH), are based on the stabilization of radicals through delocalization over the chalcone scaffolds. The presence of hydroxyl groups, which provide sources of labile hydrogen and increase the stability of the formed radicals, is responsible for antioxidant activity [178]. On the other hand, the direct pro-oxidant properties of chalcones are based on the formation of labile aroxil radicals or labile redox complexes with metal cations. It is also well known that superoxide anion O_2_^·−^ is formed during reactions between aroxil radicals and oxygen. Moreover, these radicals can form ternary compounds with DNA, copper and flavonoids, including chalcones [179]. The second mechanism of the pro-oxidant activity of chalcones is the generation of hydroxyl radicals through Fenton and Fenton-like reactions with metal ions, such as copper and iron, which accumulate in higher levels in cancer cells due to the overexpression of the transferrin receptor and copper transporter 1 [180]. This fact is in consideration of the selective cytotoxicity of polyphenols that target cancer cells. Finally, polyphenols are able to increase the accumulation of ROS through the induction of apoptosis in the early phases [181]. Polyphenol/chalcone-induced apoptosis is connected to ROS via an initiation impulse and ROS production is a consequence of apoptosis.

In the light of these facts, several natural and synthesized chalcones possess pro-oxidant activity, which is connected to their pro-apoptotic and cytotoxic potential in the treatment of gastric cancer. ROS production has been observed in human gastric cell line SGC-7901 after treatment with the ISL analog ISL-17 [52] and in MGC-803 and SGC-7901 cells following treatment with the novel quinoline–chalcone derivative 12e [49]. Fu et al. presented indirect evidence that ROS production was possibly associated with cytotoxicity in MGC-803 cells after (E)-3-(3,5-difluorophenyl)-1-(2,4,6-trimethoxyphenyl)-prop-2-en-1-one (chalcone 5) treatment [182]. In their experiment, the use of NAC (N-acetyl-L-cysteine) antioxidant reversed the cytotoxic effects and increased the viability of MGC-803 cells after chalcone 5 treatment. Moreover, NAC prevented the chalcone-induced modulation of Bcl-2, Bcl-xl and Bid, suggesting that the generation of ROS is required for chalcone 5-induced apoptosis. Similarly, the new brominated chalcone derivative H72 has been shown to induce ROS production in MGC-803 cells while NAC pre-treatment almost completely reversed H72-induced cell inhibition and apoptosis [183]. In addition, NAC has also been shown to reverse the effects of H72 on Bcl-xL, Bid, XIAP, survivin, DR4 and DR5 (TRAIL binding death receptors) expression, as well as cleaving caspase-3, -9 and PARP and changing the mitochondrial membrane potential (MMP). Zhang et al. [184] also showed that the chalcone-induced generation of ROS in MGC-803 cells modulated the activation of the keap1/Nrf2 pathway. Their novel chalcone derivative of flavokawain A (S17) induced cell inhibition in three gastric cancer cell lines (MGC-803, HGC-27 and SGC-7901), which was almost completely attenuated by pre-treatment with NAC. In the MGC-803 cells, NAC completely attenuated apoptosis by preventing the S17-induced modulation of the Bcl-2 and IAP families, the cleavage of caspases and PARP, the up-regulation of DR5 and the decrease in MMP. Furthermore, pre-treating MGC-803 with different functional ROS inhibitors, including apocynin (APO), butyIhydroxyanisole (BHA), catalase (CAT), GKT137831 (GKT), neohesperidin (NEO) and rotenone (ROT), showed that S17 produced cytotoxic ROS. These results showed that BHA and CAT significantly attenuated S17-induced cytotoxicity compared to other ROS inhibitors. The MGC-803 xenograft mouse model also demonstrated that S17 treatment led to an increase in the expression of cleaved caspase-3 and -8, as well as ROS production that was connected with S17 anti-tumor potential [184].

Both natural chalcone flavokawain B (FKB) isolated from *Alpinia pricei* Hayata and that from commercial production have shown great potential in gastric cancer treatment. Kuo et al. [185] described FKB-mediated ROS production in HCT116 cells that was attenuated by NAC. Moreover, the NAC pre-treatment abrogated the up-regulation of GADD153 (growth arrest and DNA damage-inducible gene 153) and the concurrent cleavage of PARP that was induced by FKB, suggesting that ROS accumulation is required for FKB-induced apoptosis and GADD153 expression in HCT116 cells. In addition, the use of commercial FKB and doxorubicin combination treatment for AGS gastric cancer cells has shown synergistic effects, including increased ROS generation and autophagic cell death [186]. Data have also revealed that for AGS cells, NAC pre-treatment significantly downregulates FKB–doxorubicin-mediated LC-3 I/II expression, caspase-3 activation and PARP cleavage. Moreover, it has been reported that ROS play crucial roles in ATG4B regulation that are necessary for autophagosome formation when decreased levels of ROS result in defective autophagosome assembly. The high levels of ROS that are induced by FKB–doxorubicin has been shown to reduce ATG4B activity and increase autophagic efficiency. Chang et al. [53] also showed that ROS inhibition from NAC treatment diminished FKB-induced autophagic cell death, LC3 conversion, AVO (acidic vesicular organelle) formation, p62/SQSTM1 activation, ATG4B inhibition and Beclin-1/Bcl-2 dysregulation in AGS cells. Furthermore, the results showed that FKB-induced ROS generation triggered the activation of the ERK and JNK signaling pathways and thus, mediated cell death. The use of ROS scavengers (e.g., catalase, vitamin C and Trolox) has demonstrated that ROS play important roles in the FKB-induced inhibition of AGS cell growth. These results also confirmed the effects of another flavokawain molecule isolated from *Piper methysticum* Forst roots (FKC) on HT-29 and HCT116 colon cancer cell lines [187]. FKC treatment has been shown to significantly reduce SOD antioxidant defense system activity and increase ROS production in a dose-dependent manner. Similarly, decreased SOD activity and increased ROS production have also been observed in AGS cells after Xn treatment in a time- and dose-dependent manner [188]. In addition, Xn has been found to inhibit NF-κB activity by suppressing IκBα degradation and p65 nuclear translocation, which could be reversed via ROS scavenging by NAC, suggesting that Xn alters the ROS-mediated NF-κB signaling pathway.

ROS are cell death mediators and inductors and are also involved in natural chalcone cardamonin-TRAIL receptor cell death. Cardamonin has been shown to induce ROS generation in a dose-dependent manner, which can be diminished using NAC and leads to the downregulation of the cardamonin-induced up-regulation of DR4 and DR5 expression. Moreover, NAC has been found to reverse the effects of cardamonin on the cleavage of procaspases and PARP and thus, protects cells against TRAIL-induced cell death. Cardamonin-mediated cell death has been associated with ROS production, which could also confirm that cardamonin triggers glutathione peroxidase 2 (GPx2) and thioredoxin reductase 1 (TrxR1) gene expression in Caco-2 cells, even when their activity and protein levels are not elevated [189].

In one study, two synthetic chalcones and one natural chalcone were tested on HCT116 colon adenocarcinoma cells for their pro-apoptotic and pro-oxidant activity. The natural LiA isolated from *Glycyrrhiza uralensis* induced ROS accumulation in the HCT116 cells in a time-dependent manner [190]. In this case, the addition of NAC inhibited the LiA-induced ROS production and resulted in G0/G1 phase arrest and apoptosis attenuation, along with thioredoxin (Trx) system alteration. The LiA-induced TrxR1 inhibition promoted ROS production, ASK1 expression and apoptosis in the HCT116 cells. In terms of the synthetic derivatives, the chalcone 1C showed great potential as an anti-cancer and pro-apoptotic substance. Its antiproliferative and pro-apoptotic potential was associated with oxidative stress. Additionally, Takac et al. [191] reported 1C-mediated reactive oxygen species (superoxide and lipoperoxide) and nitrogen species (RNS) generation, followed by DNA damage, cell cycle arrest at G2/M and the induction of the mitochondrial apoptotic pathway. These data were supported by the fact that NAC pre-treatment partially reversed all of the mentioned 1C-induced effects. The intracellular antioxidant defense mechanism after 1C treatment was also evaluated and showed the biphasic effect of 1C on GSH levels, with a significant decrease after 24 h of incubation along with increased GR activity. Moreover, an increase in GPx activity as the result of lipid peroxide overproduction was also observed after 1C treatment. On the other hand, the co-treatment of HCT116 cells with NAC and 1C led to the decreased phosphorylation of ROS-associated MAPK kinases (p38 MAPK, ERK1/2 and JNK), which was reflected in the partially suppressed activation of apoptosis machinery after 1C treatment, suggesting ROS involvement. Similarly, the flavone–chalcone hybrid compound 3 has been shown to induce ROS generation, with a concomitant decrease in GSH levels [192]. In silico experiments have also confirmed that compound 3 reacts with glutathione and enhances ROS by removing glutathione as the result of a sufficiently close distance between glutathione bound to GST. Compound 3-induced ROS have been found to trigger caspase-mediated apoptosis in HCT116 cells.

The upregulation of ROS generation was also observed in gastric cancer MGC-803 and SGC-7901 cell lines after treatment with a 1,2,4-triazine-chalcone derivative (9l). Moreover, ROS generation mediated by 9l has been shown to inhibit ERK pathway signaling and induce the extrinsic DR5 apoptotic pathway in gastric cancer cells [193].

The published data have clearly suggested that chalcone-mediated cell death is closely related to their chemical structure and their anti-/pro-oxidant balance, which affects their redox status, antioxidant defense mechanisms and ROS-mediated signaling pathways in gastric cancer cells.

Molecular and cellular mechanisms of the antiproliferative action of chalcones are listed in Table 1.

### 2.7. Chalcones and Multidrug Resistance

Multidrug resistance (MDR) caused by efflux membrane proteins from the ATP-binding cassette transporter superfamily (ABC transporters) is considered to be one of the potential mechanisms that leads to insufficient therapeutic outcomes [195,196,197].

Previous research has indicated that MDR is mainly the result of the action of ATP-binding cassette subfamily B member 1 protein (ABCB1), also known as P-glycoprotein (P-gp) or multidrug resistance protein 1 (MDR1). The literature has also shown the elevated expression of this transporter in both colorectal cancer [198] and gastric cancer [199]. Additionally, several studies have demonstrated the association between ABCB1 overexpression and resistant phenotypes in such tumors [200,201]. Conversely, the downregulation of ABCB1 in cancer cells leads to increased chemosensitivity to 5-FU, epirubicin, irinotecan and oxaliplatin [202,203,204,205,206], drugs that are still commonly used in the treatment of gastrointestinal carcinomas [207,208].

Currently, attention is also being paid to the identification of suitable inhibitors of efflux transporters. Due to the poor efficacy, high risk of side effects and undesirable pharmacological interactions of the already known ABC inhibitors, natural substances have become a subject of interest [209,210,211]. Likewise, several chalcones have exhibited the potential to modulate efflux transporters [212,213,214,215,216].

Several studies have also been conducted to determine the impact of chalcones on ABCB1, as well as the reversal of drug resistance in colon cancer.

A promising molecule that may be able to inhibit the efflux activity of ABCB1 is the synthetic acridine chalcone 1C [217]. This compound has exhibited antiproliferative and cytotoxic effects in ABCB1-overexpressing human colon adenocarcinoma cells (COLO 320). The compound has also been found to be able to modulate the action of the ABCB1 protein, causing the intracellular accumulation of the substrate. The 1C compound has also demonstrated a reduced antiproliferative action in non-cancer cells, indicating its relative safety. Moreover, the synergistic interaction between 1C and doxorubicin has been demonstrated by some authors.

In the same way, the synthetically prepared flavokawain B (a chalcone that naturally occurs in the kava plant) has shown cytotoxic and antiproliferative effects on different doxorubicin-resistant human colorectal adenocarcinoma cells (LoVo/Dx) [218]. Previously, the occurrence of the ABCB1 protein has been confirmed in this cell line using mRNA expression analysis, Western blots, flow cytometry, immunochemistry or functional tests [219]. Interestingly, the chemical structures of chalcones appear to be responsible for their anticancer activity as 5,7-dimethoxyflavanone (the cyclized form of flavokawain B) has only shown anticancer activity at higher concentrations compared to the effective concentrations of its parent chalcone [218]. The exact mechanism of action of flavokawain B on the ABCB1 transporter present in LoVo/Dx was not evaluated in that study; however, the authors found that the tested compounds had no impact on the cytotoxic action of doxorubicin.

Further research has revealed that chalcones may inhibit the ABCB1 function by disrupting the integration or function of cell membranes. For example, IBC has been shown to exhibit interaction potential with membranes [220]. It has also been found to elevate the intracellular accumulation of doxorubicin in doxorubicin-resistant human adenocarcinoma colon cancer cells (HT29/Dx), although the effect was milder than that of verapamil. Those authors assumed that the substance acts more as a competitive inhibitor than a displacer of substrates with lower affinity to the binding sites. However, the compound has not demonstrated significant intrinsic cytotoxic effects on any of the tested human colon cancer cells and has not been found to sensitize cells to doxorubicin, indicating that efflux by ABCB1 is not the only mechanism of resistance in these cells.

Complementary to this, human colon cancer cell lines may also express other efflux transporters, such as ABCG2 (breast cancer resistance protein, BCRP), ABCC3 (multidrug resistance-associated protein 3, MRP3) and ABCC2 (multidrug resistance-associated protein, MRP2) [221,222]. Previously, the literature has reported that some chalcones may exert a dual inhibition of ABC transporters. The synthetic chalcone CYB-2 has been shown to reverse chemoresistance and reduce the efflux activity of both ABCB1 and ABCG2 transporters, primarily through ATPase inhibition [223]. Human colon adenocarcinoma (S1-M1-80) has been used as an example of line overexpressing ABCG2 in such experiments. Chalcone CYB-2 has also exhibited cytotoxic and reversal activities in these cancer cells. Additionally, it has been suggested that mainly non-basic methoxychalcones show efficacy in the dual inhibition of efflux transporters. In brief, ABCG2 is another important ABC efflux pump that is responsible for the decreased accumulation of drugs in cells. It is also expressed in gastrointestinal malignancies and may be responsible for resistance to irinotecan, which is a drug used in the treatment of several gastrointestinal malignancies [224]. Moreover, LiA has been shown to be a compound that is able to reverse chemosensitivity to mitoxantrone and topotecan, as well as the inhibition of ABCG2 efflux activity in the S1-M1-80 cell line [225]. These effects could be explained by the ability of licochalcone to bind to the substrate-binding pocket of ABCG2. Based on the comparable cytotoxic activity of LiA in both the parent and ABCG2 overexpressing cell lines, it has been assumed that the of efflux of the tested chalcone using the mentioned transporter is not possible. In contrast, the combretastatin A4 analogous chalcone and its Pt-complex have not displayed significant efficacy against overexpressing MRP3 or MRP1 in the human colon adenocarcinoma cell line HT-29, depending on the stimulation of cells by colchicine [226]. Experiments on other cell lines have revealed that these two compounds could be substrates for ABC transporters. In earlier research, Xn (a major prenylated chalcone derived from the female flowers of common hop, *Humulus lupulus*) was shown to exhibit cytotoxic activity in the human colon adenocarcinoma cell line HCT-15 [105]. Additionally, evidence has pointed to the endogenous presence of the ABCB1 transporter in this cell line [227].

On the other hand, the findings mentioned above should be interpreted with caution because of the unclear role of transporter inhibition in the development of gastrointestinal malignancies. Surprisingly, some authors have reported lower ABCB1 mRNA expression in the early stages of colorectal cancer (adenomas and carcinomas) compared to that in morphologically normal tissues from the same patient [228]. Furthermore, some substrates and competitive inhibitors of ABCB1 have also been found to act as inducers of the expression of the respective transporter in some cells [229,230,231].

Less is known about the ability of chalcones to influence multidrug resistance that is caused by membrane ABCB1 efflux pump in gastric cancer.

A study using an Rh123 uptake assay demonstrated that cardamonin, a natural chalcone, was able to reduce ABCB1 efflux activity in 5-FU-resistant gastric cells (BGC-823/5-FU) [73]. Moreover, a concentration-dependent relationship was confirmed. It has also been suggested that the downregulation of ABCB1 could be the result of the inhibition of the Wnt/ß-catenin signaling pathway. This assumption is based on the fact that ABCB1 is under the control of this pathway [232,233]. The effect of the tested compound on the efflux ABCB1 activity in 5-FU-sensitive cancer cells (BGC-823) was not obvious. Hou et al. [73] also observed the synergistic action of the cardamonin and 5-FU combination in the reduction in the mRNA and protein expression of ABCB1. In vivo experiments on xenograft mouse models have also shown the increased inhibition of tumor growth when this combination of compounds is used compared to monotherapy with cardamonin or 5-FU. To summarize, chalcones affect ABC transporters depending on their structures and doses, as well as the type of cell line, showing their different mechanisms of action.

## 3. Animal Studies

The above-mentioned in vitro studies indicated several modes of antiproliferative action among chalcones in colorectal cancer. On the other hand, the in vivo anticancer effects of chalcones remain to be fully elucidated.

### 3.1. Chemically Induced Colon Cancers

Azoxymethane-induced colon cancer is a commonly used model in the study of the chemoprevention or treatment of colon cancer.

Kim et al. [234] studied the anticancer and antimetastatic effects of LiA in azoxymethane-treated male C57BL/6 mice. The oral administration of LiA suppressed carcinogenesis in a dose-dependent manner, as evidenced by reductions in tumor incidence and the number of tumors per animal. Immunohistochemical analyzes showed decreases in the markers associated with either cell proliferation or inflammation in the LiA-treated mice. Moreover, LiA also reduced the liver metastasis that was induced by the intrasplenic administration of CT-26 cells.

Recently, Liu et al. [235] documented the ability of Xn to decrease the incidence of azoxymethane-induced aberrant crypt foci (ACF), which is a biomarker for CRC. The oral administration of Xn (5 mg/kg) to male Sprague Dawley rats for 8 weeks resulted in significant decreases in the numbers of ACF/colon compared to the azoxymethane-treated group. Simultaneously, Xn also suppressed several markers of oxidative stress and inflammation, including malondialdehyde, inducible nitric oxide synthase and COX-2.

In addition, the pro-carcinogenic effect of azoxymethane has also been found to be suppressed by the chalcone from the roots of licorice (ISL). Baba et al. [236] studied the chemopreventive effect of ISL using ddY mice after the subcutaneous administration of azoxymethane. As previously documented, the administration of ISL in diets significantly suppresses ACF formation, as well as the formation of colon cancers. Later, Takahashi et al. [97] investigated the effect of ISL on azoxymethane-treated F344 rats. The peroral administration of ISL significantly decreased the number and size of ACF. Isoliquiritigenin has also demonstrated an excellent chemopreventive effect in azoxymethane and dextran sodium sulfate (AOM/DSS) model of colon carcinogenesis in male BALB/c mice. The oral administration of ISL over 12 weeks decreased the incidence, size and multiplicity of colon adenomas in a dose-dependent manner. Additional analyses have shown that the protective effect of ISL is associated with the inhibition of M2 macrophage polarization. It seems that this effect can also be accompanied by the suppression of PGE2 and IL-6 signaling [237]. Additionally, Wu et al. [156] confirmed the protective effect of ISL in the same colon cancer model. The application of ISL decreased the incidence of colon tumors in a dose-dependent manner. Moreover, ISL significantly decreased the levels of different cytokines, including IL-6, IL-10, TNF-α and IL-1β. In addition, the effect of ISL on gut microbiota could play a role in its protective action. Isoliquiritigenin has been found to restore the dysregulated composition of gut microbiota after the application of AOM/DSS, suppress opportunistic pathogens and elevate commensal bacteria, including *Prevotella, Butyricicoccus, Clostridium* and *Ruminococcus*.

A similar chemopreventive effect has also been described for cardamonin in an AOM/DSS model of colon carcinogenesis. Cardamonin, a natural chalcone from the Zingiberaceae family, has been found to reduce the number and size of tumors compared to those in control groups. Histopathological analyzes have shown that it can significantly reduce mucosal infiltration with inflammatory cells, as well as the presence of dilated goblet cells in cardamonin-treated cells, which indicates that this chalcone could protect against colitis-associated colorectal cancer. Moreover, cardamonin has been found to alter the expression of several miRNAs in experimental animals [238]. Some authors have also documented different levels of miRNA expression between vehicle- and cardamonin-treated animals [77]. As previously suggested, the modulation of miRNAs could be associated with redox signaling alterations in cardamonin-treated animals and the subsequent modulation of oxidative stress.

Furthermore, 1,2-dimethylhydrazine (DMH), a strong DNA alkylating agent, is another inducer of colon cancer. Pande et al. [239] evaluated the anticancer effects of new 2′-hydroxy chalcone derivatives in male Wistar rats. Of the six chalcones studied, only compound C1 (1-(2-Hydroxy-phenyl)-3-p-tolyl-propenone) significantly reduced both the incidence of ACF and the number of colon adenocarcinomas. Moreover, this chalcone significantly reduced TNF-α concentrations in colon homogenates. Later, Jin et al. [240] evaluated the chemotherapeutic activity of the ruthenium-phloretin complex (RPC) using a DMH/DSS model in male Wistar rats. The chalcone complex RPC significantly reduced the incidence of ACF, as well as hyperplastic lesions. Moreover, it increased the levels of antioxidant enzymes and glutathione, indicating the role of free radicals in apoptosis.

### 3.2. Xenograft Models of Colon Carcinogenesis

The protective effects of chalcones have also been studied in xenograft models of colorectal carcinogenesis. Hayashi et al. [241] evaluated the anticancer effect of a new quercetin chalcone hybrid. The oral administration of the quercetin chalcone hybrid reduced the size and weight of colon-25 tumors that were implanted in BALB/c mice.

The above-mentioned chalcone (ISL) has also been found to significantly reduce tumor growth in xenograft-induced colon cancer. In one study, the application of HCT-116 cells to male BALB/c mice resulted in colon carcinogenesis and the development of colon cancer. Then, the application of ISL for 14 days after tumor development led to the significant suppression of tumor growth. Further TUNEL and immunohistochemical analyzes revealed a significant increase in the expression of p62/SQSTM1 (a possible regulator of apoptosis) and cleaved caspase-8 in the tumor tissues [242]. Another well-known natural chalcone (LiA) has been shown to significantly inhibit the growth and reduce the size of tumors in CT-26 cell-inoculated mice. Moreover, in combination with cisplatin, ISL has been proven to significantly reduce cisplatin-induced nephrotoxicity and hepatotoxicity, as evidenced by decreases in serum blood urea nitrogen, malondialdehyde levels and liver aminotransferases, such as ALT and AST [243]. Similar results have been obtained in other studies on ISL by the same authors. Isoliquiritigenin, similar to LiA, has been shown to reduce the size of tumors in the same model. In addition, ISL has been found to suppress the nephro- and hepatotoxicity of co-administrated cisplatin without decreasing its therapeutic effect [244]. Additionally, Kim et al. [245] demonstrated the anticancer effect of LiA in BALB/c mice with CT-26 colon cancer. Licochalcone A significantly decreased tumor sizes in comparison to those in the untreated control group. Subsequent immunohistochemical analyzes showed a decrease in CD31 (a specific endothelial cell marker), as well as proliferation marker Ki-67 levels.

In addition, Yin et al. [246] recently documented the ability of chalcones to suppress colon carcinogenesis in a patient-derived xenograft model. In mice treated with the chalcone derivative HCI-48 (1.5 or 3 mg/kg), tumor growth was significantly reduced in both experimental groups in comparison to a vehicle-treated group. Their immunohistochemical analyzes also revealed a significant decrease in the expression of the proliferation marker Ki-67, as well as the phosphorylation of several proteins, including BAD, STAT-3, AKT and EGFR1.

Huang et al. [247] reported the anticancer effect of butein in HCT116-bearing BALB mice. In butein-treated animals, tumor size was significantly smaller compared to that in untreated animals. Their in vitro analysis showed the ability of butein to suppress securin expression in human colorectal cells, which could also be involved in the anticancer effect of butein in vivo.

### 3.3. Chalcones and Gastric Cancer

Furthermore, the anticancer effects of chalcones have also been studied in gastric cancer models. Flawokawain B has been shown to effectively suppress tumor growth in AGS-xenografted mice. Western blot analyzes of tumor sections have shown increases in the levels of autophagy markers [53]. A recent study by Hou et al. [73] reported the anticancer effect of cardamonin in a xenograft mouse model. Cardamonin suppressed tumor growth in mice bearing BGC-823 gastric cancer cells that were resistant to 5-FU; however, the combination of cardamonin and 5-FU resulted in a more significant anticancer effect in comparison to either cardamonin or 5-FU alone. Another recent study demonstrated suppressed gastric tumor growth in MGC-803 xenografted mice that were treated with a novel triazine–chalcone derivative. Tumor tissue analyzes showed the down-regulation of Ki67 and p-ERK and increased levels of cleaved PARP. Moreover, the anticancer effect of the most active chalcone was superior to that of 5-FU [193]. A similar tumor-suppressive effect of chalcones was documented by Zhang et al. in the same model of carcinogenesis [183]. Another study showed that the intraperitoneal application of a newly synthesized brominated chalcone (H72) led to a decrease in tumor weight without any signs of toxicity. This compound also increased ROS production and modulated Bcl-2 family protein activity under in vitro conditions. Furthermore, Dong et al. [98] found that a new benzochalcone derivative (KL-6) remarkably decreased tumor growth in BALA/c nude mice with MKN1 gastric cancer cells. In addition, this chalcone also decreased the number of metastatic lesions in the livers of the mice. The antitumor effect of selected chalcones is summarized in Table 2.

### 3.4. Chalcones and Helicobacter pylori Infection

The bacterium *Helicobacter pylori* (*H. pylori*) was first isolated in 1982 from the cultures of endoscopic biopsy specimens from patients with gastritis and peptic ulceration. For this historic discovery, Marshall and Warren were awarded the Nobel Prize in Physiology or Medicine in 2005 [250]. In 1994, the International Agency for Research on Cancer, the specialized cancer agency in the WHO, identified the bacterium *H. pylori* as a type I carcinogen. Since then, it has been proven that this spiral, microaerophilic and Gram-negative bacterium is able to colonize the human gastric epithelium with a prevalence of 85–95% in developing countries and 30–50% in developed countries [251]. *H. pylori* infection is clinically manifested as dyspepsia, chronic gastritis, duodenal and gastric ulcers, gastric adenocarcinoma and MALT lymphoma (gastric mucosa-associated lymphoid tissue lymphoma) [252]. Together with host factors and other risk factors, such as high salt consumption and smoking, it is involved in the multistep conversion of healthy gastric mucosal tissues into chronic superficial gastritis, followed by atrophic gastritis and intestinal metaplasia, which ultimately leads to dysplasia and adenocarcinoma. Inflammation and changes in gastric epithelial cells are responses to the direct action of various bacterial virulence factors (pathogen-associated molecular patterns, PAMPs) that develop after *H. pylori* infection. Such factors include cagPAI (cag pathogenicity island) encoding CagA protein, peptidoglycan, VacA toxin, adhesive molecules, OMPs (outer membrane proteins), urease and others [253,254].

In this context, *H. pylori* eradication may represent a key step in the prevention and treatment of *H. pylori*-positive gastric cancer. Chalcones are undoubtedly potent natural substances with antibacterial activity against *H. pylori*. They are known for their multiple effects, including antibacterial, antiproliferative and antitumor activities [255]. Chalcones have been found to show specific activity against the substances produced by *H. pylori*, signaling pathways that are affected by the presence of *H. pylori* and thus have a gastroprotective effect (Figure 4). Xinjiachalcone A is isolated from the roots of *Glycyrrhiza inflata* Batalin (Fabaceae) and has shown strong antibacterial activity against 17 strains of *H. pylori* (commercially available and clinically isolated strains associated with gastritis, duodenal ulcer and MALT). The minimum inhibitory concentration (MIC) has varied between 12.5 and 50 µM under in vitro conditions [256]. In addition to low MIC and strong antibacterial activity, chalcones have also inhibited interactions between *H. pylori* and the epithelium and its motility. The urease- and flagella-mediated motility of *H. pylori* is an essential process for gastric epithelium colonization. Therefore, Yoshiyama et al. [257] studied the effect of sofalcone [(2′-carboxymethyl 4,4′-bis(3-methyl-2-butenyloxy)chalcone)] on the chemotactic motility of *H. pylori*. Sofalcone, a derivative of sophoradin that is isolated from the root of the Chinese plant *Sophora subprostrata*, is currently used as a gastric mucosa protective agent in Japan [258]. At a concentration of 222 µM, this chalcone has been shown to inhibit swarming zone formation and reduce bacterial chemotaxis by 90% compared to untreated cells. These effects have been associated with the inhibition of urease activity [257]. The cytoprotective and anti-inflammatory effects of sofalcone have also been associated with the modulation of several intracellular pathways, including the activation of the transcription factor Nrf2 (nuclear factor erythroid 2–related factor 2), which activates the expression of numerous cytoprotective genes after translocation into nuclei, such as HO-1 (Heme oxygenase-1). Its natural repressor is Kelch-like ECH-associated protein 1 (KEAP1), which ensures the cytosolic sequestration and proteasomal degradation of Nrf2. Sofalcone activates the Nrf2-HO-1 pathway in human colon carcinoma cells by increasing the accumulation of Nrf2, resulting in the induction of HO-1, which is responsible for the response to oxidative stress. This effect is mediated by the covalent binding of sofalcone to Keap1, which reduces its interaction with Nrf2. The induction of this pathway is considered to be a therapeutic strategy for the treatment of colitis, which has also been proven by the results of experiments on rats. After rectal administration, sofalcone has been found to reduce colonic inflammation and damage, myeloperoxidase activity and the levels of pro-inflammatory factors, such as CINC-3, iNOS and COX-2 [259].

Another pathway that is considered to be a key signaling pathway mediating *H. pylori*-associated gastric inflammation is the NF-κB pathway. NF-κB is an inducible transcription factor that modulates the expression of various inflammatory substances, such as cytokines, chemokines and adhesion molecules, and it also regulates cell proliferation, differentiation and apoptosis [260]. For this reason, the suppression of this pathway, as a pro-inflammatory gene inducer, is considered to be a therapeutic strategy in the treatment of chronic gastritis. Within this context, chalcones have been proven to be effective both in vitro and in vivo. In recent years, several chalcones capable of inhibiting the inflammatory response induced by *H. pylori* have been synthesized and studied for this purpose. Some of the chalcone analogs have shown that in addition to inhibiting the adhesion and invasion of *H. pylori* to gastric epithelial cells (AGS cells), they are able to significantly inhibit NF-κB activation and the NF-κB-dependent expression of IL-8 (at concentrations of 2.5, 5 and 10 µM). Paradoxically, three antimicrobial agents that are widely used to eradicate *H. pylori* have shown no effects on this pathway (at a concentration of 2.5 µM) [261]. Authors who have studied extracts containing naturally occurring chalcones, such as xanthohumol A and D, have obtained similar results. The extracts have been found to inhibit Nf-kB-driven transcription and nuclear translocation, as well as TNF-α-induced IL-8 secretion, in a concentration-dependent manner. The authors have mainly attributed these effects to hydroalcoholic extracts and isolated chalcones, which have been identified as the main components of the extracts [262].

The combination of the PAMPs and PRRs (pattern recognition receptors) of hosts through the Nf-kB pathway promotes the expression of pro-inflammatory cytokines and activates immune response, including the formation of inflammasomes. Inflammasomes are large cytosolic protein complexes, the activation of which leads to the maturation and secretion of cytokines, such as IL-1β (considered to be a pro-inflammatory cytokine that plays a central role in gastric tumorigenesis) and IL-18. In the context of *H. pylori* infection, the NLRP3 inflammasome ((NOD)-like receptor family, pyrin domain-containing 3) has been the most studied. It facilitates the infiltration of neutrophils and reductions in gastric acid production, supports the persistence of bacteria in the stomach and contributes to the development of chronic inflammation [263]. The results of one study on the human monocytic THP-1 cell line demonstrated the remarkable anti-inflammatory properties of several chalcone derivatives. For example, two of the five tested substances inhibited the *H. pylori*-induced production of IL-1β and IL-18 in these cells at a concentration of 10 µM. The secretion of these cytokines depends on the activation of caspase-1, which is involved in the cleavage and maturation of IL-1β (also considered to be a marker of the activation of the NLRP3 inflammasome) and IL-18. The selected chalcones in that study inhibited the activation of the NLRP3 inflammasome by blocking the oligomerization of ASC (the apoptosis-associated speck-like protein that contains a caspase recruitment domain), which is a direct indicator of the activation of NLRP3 inflammasome and the cleavage of the pro-forms of caspase-1, IL-1β and IL-18 into their active forms without changing their expression. The analyzes of their effects on IRAK4, IkBα, NF-κB and proteins from the MAPK family also helped to clarify the mechanisms of action of these chalcones. The protein kinase IRAK4 mediates the signaling of immune reactions from toll-like receptors, leading to the activation of IKB kinases (IKK) and the subsequent phosphorylation and degradation of IkBα (a negative regulator of NF-κB). In *H.pylori*-infected cells, exposure to chalcones significantly suppresses IRAK4 activation, reduces IkBα phosphorylation and the translocation of NF-κB to nuclei and reduces active caspase-4 levels without affecting the phosphorylation status of p38 MAPK and ERK. These results have indicated that the studied chalcones inhibit the canonical and non-canonical pathways of the NLRP3 inflammasome [264]. The naturally occurring chalcone veluton F and some of its regioisomers have also exhibited NLRP3 inflammasome inhibitory activity by inhibiting the release of IL-1β in both in vitro cell systems and in vivo animal models [265].

Oxidative stress also contributes to the process of inflammatory response. It is well known that *H. pylori* infection is accompanied by the formation of reactive oxygen and nitrogen species and prostaglandin E2 (PGE2), which are produced by NADPH oxidase, inducible nitric oxide synthase (iNOS) and COX-2 [266]. Natural chalcones isolated from *Angelica keiskei*, have been shown to reduce NO production in LPS-activated macrophages (RAW 264.7 cells) at concentrations of less than 5 µM, in addition to inhibiting IL-6 and IL-1β expression, degrading IκBα and translocating NF-kB (p65 subunit) into nuclei. They have also been found to suppress the expression of COX-2 and iNOS at the protein and mRNA levels [267]. Other synthetic and natural chalcone derivatives have also been observed acting similarly. Phenyl-thiophenyl propenone inhibits *H. pylori*-induced increases in intracellular ROS levels in gastric epithelial cells (AGS cells), induces the suppression of the activated p38 MAPK, ERK and JNK signaling pathways and reduces IL-8 expression at the mRNA and protein levels [268]. In one study, an extract with a high content of natural chalcones (primarily LiA) was used in a non-cytotoxic concentration and proved to be an effective scavenger of ROS in *H. pylori*-infected gastric epithelial cells (AGS cells). In another study, an S-lico extract reduced the expression of inflammatory mediators, such as COX-2, iNOS, TNF-α and VEGF (a key growth factor that is essential for angiogenesis, solid tumor growth and metastasis). Its anti-inflammatory and anti-cancer effects were also demonstrated in vivo as the pathological changes (inflammation, mucosal erosion, dysplasia and gastric adenoma) induced by *H. pylori* were reduced after administering 50 and 100 mg/kg of the extract to mice [248]. Other plant extracts that are rich in polyphenols and chalcones have also shown satisfactory effects in animal models. In one study, the plant leaf extract (LLF) from *Nelumbo nucifera* Gaertn (Indian lotus), which contains flavonoids, such as kaempferitrin, hypericin, astragalin, phlorizin and quercetin, was administered to mice infected with *H. pylori*. At a dose of 100 mg/kg/day for 2 weeks, there was a significant reduction in gastric lesions compared to both the control and ranitidine groups. The authors also noted significant decreases in the inflammatory cytokines IL-6, IL-12, TNF-α and IFN-γ and changes in the levels of substance P, endothelin-1, vasoactive intestinal polypeptide (VIP) and somatostatin in the murine serum after the administration of the LLF [249].

These experimental results suggest that chalcones and their derivatives could be important molecules in the prevention and treatment of infections and *H. pylori*-induced gastritis, as well as *H. pylori*-associated cancers.

## 4. Conclusions

The chalcones attracted the attention of the scientific community in the last decades not only due to their diverse biological properties and low toxicity but also due to the relatively easy synthesis of new analogs with great therapeutic potential. This review summarizes the experimental evidence about the anticancer effect of natural and synthetic chalcones focused on colorectal and gastric cancer. The chemotherapeutic effects of chalcones are mediated by numerous activities including the initiation of cell death machinery, suppression of cancer cell growth, modulation of several signaling pathways such as Wnt/β-catenin pathway, nuclear factor kappa B pathway, MAPK pathway and many others. In addition, chalcones seem to have an anticancer effect also via the inhibition of angiogenesis, production of ROS, anti-*H. pylori* activity and anti-inflammatory action. Taken together, both in vitro and experimental in vivo data have shown the potential of chalcones to treat colorectal and gastric cancer although further studies based on clinical trials are essential to validate their efficacy in anticancer therapy.

## Figures and Tables

**Figure 1 ijms-24-05964-f001:**
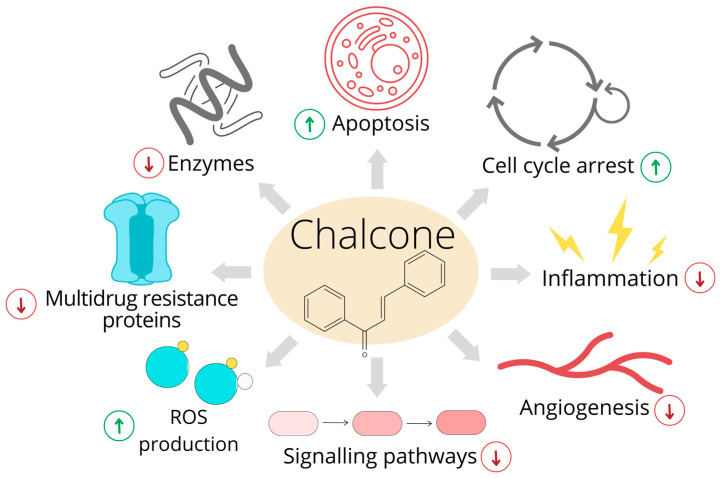
Anticancer effects of chalcones. The original figure was made for this review using the Canva software by Radka Michalkova.

**Figure 2 ijms-24-05964-f002:**
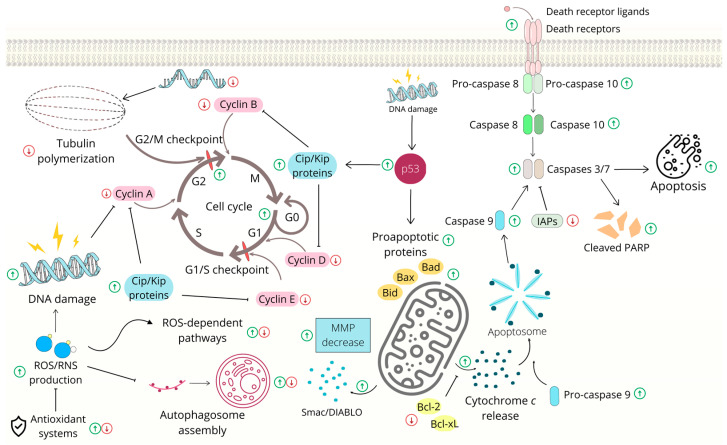
Cell cycle arrest and apoptosis induced by chalcones in gastrointestinal cancer cells. Abbreviations: (pro)Casp—(pro)caspase, Bcl-2/Bcl-Xl—antiapoptotic proteins, Bad/Bax/Bid—proapoptotic proteins, Cip/Kip—CDK interacting protein/Kinase inhibitory proteins, IAPs—inhibitors of apoptosis, MMP- mitochondrial outer membrane permeabilization, PARP—poly (ADP-ribose) polymerase, ROS/RNS—reactive oxygen species/reactive nitrogen species, Smac/DIABLO—second mitochondria-derived activator of caspase/direct inhibitor of apoptosis-binding protein with low pI. The original figure was made for this review using the Canva software by Radka Michalkova.

**Figure 3 ijms-24-05964-f003:**
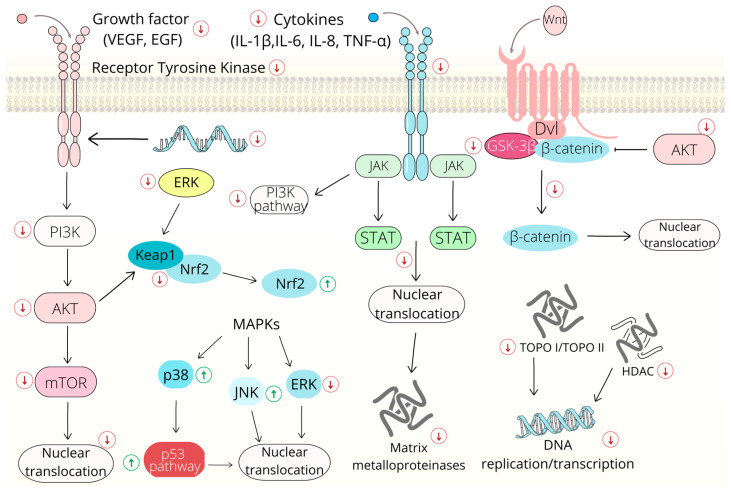
Effect of chalcones on selected signaling pathways. Mechanism of modulation of intracellular signaling pathways in gastrointestinal cancer cells. Abbreviations: AKT—protein kinase B, Dvl—disheveled, EGF—epidermal growth factor, ERK—extracellular signal-regulated kinase, GSK-3β—glycogen synthase kinase-3 beta, HDAC—histone deacetylase, IL—interleukin, JAK—Janus kinase, JNK—c-Jun N-terminal kinases, Keap1—Kelch-like ECH-associated protein 1, mTOR—mammalian target of rapamycin, Nrf2—NF-E2–related factor 2, p38—p38 mitogen-activated protein kinase, p62—sequestosome 1, PI3K—phosphoinositide 3-kinase, ROS—reactive oxygen species, STAT—signal transducer and activator of transcription, TOPO I/II—topoisomerases I/II, TNF-α—tumor necrosis factor alpha, VEGF—vascular endothelial growth factor. The original figure was made for this review using the Canva software by Radka Michalkova.

**Figure 4 ijms-24-05964-f004:**
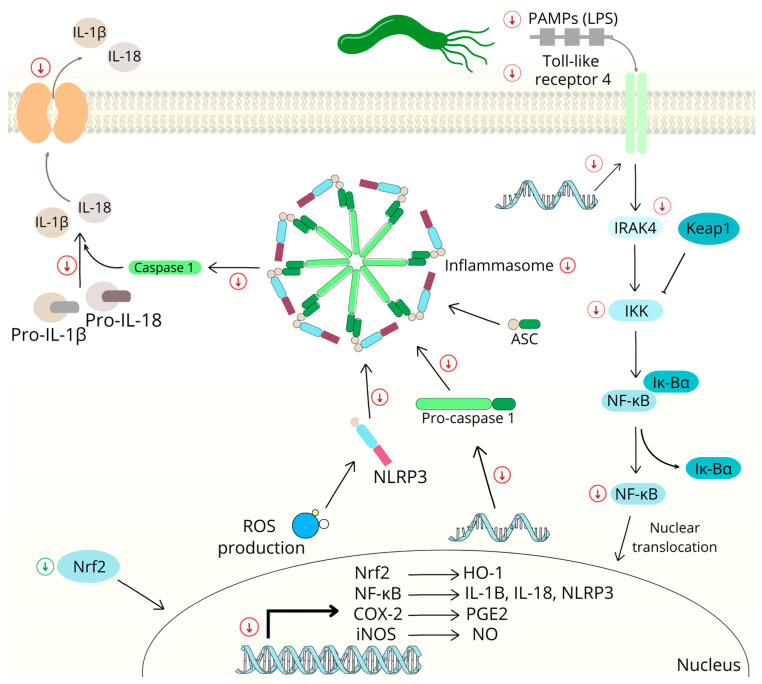
Effect of chalcones inflammation and *H. pylori* infection. Abbreviations: ASC—apoptosis-associated speck-like protein containing a CARD, COX-2—cyclooxygenase-2, HO-1—heme oxygenase 1, IL—interleukin, Iκ-Bα—nuclear factor of kappa light polypeptide gene enhancer in B-cells inhibitor alpha, IKK—IkappaB kinase, iNOS—nitric oxide synthase, IRAK4—interleukin-1 receptor-associated kinase 4, Keap1—Kelch-like ECH-associated protein 1, LPS—lipopolysaccharide, NF-κB—nuclear factor kappa-light-chain-enhancer of activated B cells, NLRP3—NLR family pyrin domain containing 3, NO—nitric oxide, Nrf2—NF-E2–related factor 2, PAMPs—pathogen associated molecular patterns, PGE2—prostaglandin E2, ROS—reactive oxygen species. Original figure made for this review using the Canva software by Radka Michalkova.

**Table 1 ijms-24-05964-t001:** Antiproliferative actions of selected chalcones based on in vitro studies.

Chalcone	IC_50_ or Used Concentration	Cell Line	Mechanism of Action	Reference
(E)-3,4-dihydroxy-2′-methoxychalcone	10 µmol/L	CT26.WT	Inhibition of proliferation, colony formation, migration and invasion, induction of apoptosis ↓ Bcl-2, phospho-Akt, IkB-α, nuclear p65 ↑ Bax, cleaved caspases—3, cleaved PARP, E-cadherin	[23]
(E)-2-(2′,4′-dimethoxybenzylidene)-1-tetralone	3.44 µmol/L	HCT116	Induction of apoptosis, G2/M phase cell cycle arrest, DNA damage and changes in microtubule structure ↑ activity of caspases—3/-7, cleaved PARP,	[25]
(E)-2-(4′-methoxybenzylidene)-1- Benzosuberone, (E)-2-(2′,4′- dimethoxybenzylidene)-1-tetralone	0.5 and 5 µmol/L	Caco-2	Cell cycle arrest in G2/M phase cell cycle arrest, induction of apoptosis, DNA fragmentation ↓ MMP, expression of Bcl-2, Bcl-xL, α, α1 and β5 tubulins ↑ activity of caspase -3, production of ROS, expression of Bax,	[26]
(2 E)-3-(acridin-9-yl)-1-(2,6-dimethoxyphenyl)prop-2-en-1-one	10 µmol/L	HCT116	Inhibition of proliferation, induction of apoptosis, G2/M cell cycle arrest, tubulin dysregulation ↓ MMP, phospho-survivin ↑ cytochrome *c*, Smac/Diablo, activity of caspases -3, -7 and -9, cleaved PARP, phospho-ATM, phospho-SMC, phospho-H2A.X, phospho-p53, p53, phospho-Bad, Bad, p21, phospho-p38 MAPK, phospho-Erk1/2, phospho-JNK, phospho-Akt	[27]
(2E)-3-(3-bromophenyl)-1-(2-hydroxyphenyl)prop-2-en-1- one, 2-(4-bromophenyl)-3-hydroxy-4H-chromen-4-one	10–15 µmol/L	HCT116	Inhibition of proliferation, induction of apoptosis, S and G2/M cell cycle arrest, nuclear condensation ↑ p53, p27, cleaved PARP	[29]
2′-hydroxy-2,4,6-trimethoxy-5′,6′-naphthochalcone	10 µmol/L	SW620	Inhibition of proliferation, colony formation, induction of apoptosis, G2/M cell cycle arrest, disruption of the microtubular network, DNA fragmentation ↓ cyclins D1, A, B1, ↑ phospho-Aurora A/B/C, activation of caspases -2, -3, -7, -9, cleaved PARP, phospho-ATM, phospho-ATR, phospho-γ-H2A.X, phospho-p53, p53, Bax, phospho-Chk2,	[30]
Xanthohumol	5–20 µmol/L	HT-29	Induction of apoptosis, G2/M cell cycle arrest ↓ Bcl-2, cyclin B1, Ras, phospho-MEK, phospho-Erk ↑ Bax, activation caspases -3, -9	[31]
(E)-2-(3-(2,5-dimethoxyphenyl)-3-oxoprop-1-enyl)quinazolin-4(3H)-one	3.56 µmol/L	HCT116	Induction of apoptosis, G2/M cell cycle arrest ↓ MMP ↑ cleaved caspases -3, -7, -9, cleaved PARP	[40]
Sappanchalcone	10–50 µmol/L	HCT116	Induction of apoptosis ↓ MMP, Bcl-2 ↑ Bax, cleaved caspases -3, -7, -8, -9, cleaved PARP, ROS production, AIF, phospho-p53, p-53	[42]
2′,4′-dihydroxychalcone	5–20 µmol/L	MGC-803	Inhibition of proliferation, nuclear condensation and ear fragmentation, G2/M cell cycle arrest ↓ expression of survivin ↑ activity of caspases -3	[50]
(E)-1-(4-((2-methylquinolin-4-yl)amino)phenyl)-3-(3,4,5-trimethoxyphenyl)prop-2-en-1-one	300–1500 nmol/L	MGC-803	Inhibition of proliferation and colony formation, G2/M cell cycle arrest ↑ cleaved caspases -3, -9, cleaved PARP, ROS production	[49]
Halogenated derivatives of chalcones	2.5–20 µmol/L	RAW 264.7 and mouse peritoneal macrophages	Activity against LPS-induced release of cytokines ↓ TNF-α, IL-1β, IL-6, IL-12 and COX-2 expression, phospho-JNK, IkB, phospho-p38, phospho-Erk, p65	[194]
Phloretin	9–36 µmol/L	SNU-1	Induction of autophagy, G0/G1 cell cycle arrest, inhibition of invasiveness and migration, ↓ cyclins D1 and D2, phospho-p38, phospho-Erk1/2 ↑ Beclin-1, LC3B	[24]
(E)-3-(3-fluoro-4-hydroxyphenyl)-1-(3,4,5-trimethoxyphenyl)prop-2-en-1-one	40 μmol/L	SGC-7901, BGC-823	Inhibition of cell proliferation and induction of apoptosis, G2/M cell cycle arrest ↓ cdc2, cyclin B1, Bcl-2, p62, phospho-Akt, phospho-mTOR ↑ cleaved PARP, Bax, ROS generation, Beclin-1, LC3A/B	[52]
Flavokawain B	2.5–10 μg/mL	AGS cells	Inhibition of cell proliferation and colony formation, induction of autophagy, G2/M cell cycle arrest ↓ phospho-mTOR, ATG4B, cyclins A, B1, Cdk1, cdc25C, phospho-HER2, HER2, phospho-PI3K, PI3K, phospho-Akt, Akt, Bax ↑ LC3A/B, p62, cleaved PARP, Beclin-1/Bcl-2 ratio, ROS production, phospho-JNK1/2, phospho-Erk1/2	[53]
(E)-4-chloro-N-(4-(3-oxo-3-(3,4,5-trimethoxyphenyl) prop-1-en-1-yl) phenyl) butanamide	1.5–6 μmol/L	MGC-803	Inhibition of proliferation and colony formation, induction of apoptosis ↓ cyclin B1, Cdk1, Bid, Bcl-2, XIAP, c-IAP1 ↑ DR5, cleaved caspases -3, -7, -8, -9, cleaved PARP, Noxa	[55]
2′-dihydroxy-4,4′-dimethoxydihydrochalcone	8-32 μmol/L	MKN45	Inhibition of cell proliferation, induction of apoptosis and autophagy, inhibition of invasiveness and migration ↓ MMP2, MMP9 ↑ ROS generation, Beclin-1, Atg6, Atg7, LC3-II, phospho-MEK, phospho-Erk	[59]
Liquiritin	80 μmol/L	SGC7901/DDP	Inhibition of proliferation and colony formation, invasiveness and migration, induction of apoptosis and autophagy, DNA damage ↓ cyclin D1, A, Cdk4, MMP, p62 ↑ p21, p53, cleaved caspases -3, -8, -9, cleaved PARP, Beclin-1, LC3I/II	[60]
Derricin and derricidin	30, 50 μmol/L	HCT116, DLD-1	Inhibition of proliferation, S and G2/M cell cycle arrest ↓ β-catenin, Wnt reporter activity	[71]
Isobavachalcone	50, 100 μmol/L	HCT116, SW480	Inhibition of proliferation and colony formation, induction of apoptosis ↓ Bcl-2, XIAP, survivin, β-catenin, phospho-GSK-3β, phospho-Akt ↑ cleaved PARP, cleaved caspase -3, Bax, phospho-β-catenin	[72]
Licochalcone A	10–50 μmol/L	HCT116, HeLa, A549, Hep3B	Inhibition of proliferation and colony formation, induction of apoptosis ↓ PD-L1, Ras, phospho-Raf, phospho-MEK, phospho-p65, phospho-IkBα, phospho-IKKα/β, TRAF2, RIP1 ↑ cleaved PARP, cleaved caspase-8	[75]
Thioderivatives of chalcones	1–5 μmol/L	HCT116, DLD-1	Inhibition of proliferation, invasiveness and migration, induction of apoptosis, S phase cell cycle arrest ↓ p50 in nuclear fraction, p65 nuclear fraction, COX-2, EGFR, phospho-Akt, STAT3, Bcl-xL, c-Myc, ↑ nuclear phospho-Nrf2, SOD, GSTP	[76]
Cardamonin	20–80 μmol/L	HCT116	Inhibition of proliferation, induction of apoptosis ↓ c-Myc, Oct4, Cyclin E, TSP50, NF-kB ↑ activity of caspases -3, -9, Bax,	[79]
Hydroxysafflor Yellow A	20 μmol/L	KYSE-30	Inhibition of proliferation, invasiveness and migration, induction of apoptosis ↓ ICAM1, VCAM1, MMP9, TNF-α, phospho-p65, phospho-IkBα	[83]
Butein	1–50 μmol/L	SAS, KB	Inhibition of proliferation, invasiveness and migration, induction of apoptosis ↓ phospho-p65, p65, COX-2, MMP-9, survivin	[84]
Flavokawain C	60 μmol/L	HCT116	Inhibition of proliferation and colony formation, invasiveness and migration, induction of apoptosis, S and G2/M cell cycle arrest, DNA damage ↓ MMP, c-IAP1, XIAP, c-FLIP_L_, survivin, Cdk2, Cdk4, phospho-Rb, Rb, phospho-Akt ↑ cleaved caspases -3, -8, -9, cleaved PARP, Smac/DIABLO, AIF and cytochrome *c* release, Bak, p21, p27, GADD153, phospho-Erk	[88]
Cardamonin	10 μmol/L	BGC-823/5- FU, BGC-823	Induction of apoptosis, inhibition of proliferation, G2/M phase cell cycle arrest ↓ function of P-gp, P-gp, TCF4, β-catenin	[73]
3-deoxysappanchalcone	5–20 μmol/L	HCT15, HCT116	Inhibition of proliferation and colony formation, G2/M cell cycle arrest ↓ cyclin B1, phospho-TOPK, TOPK, phospho-Erk, phospho-RSK, phospho-c-Jun ↑ p53, p21, cleaved PARP, cleaved caspases -3, -7	[90]
Isobavachalcone	20, 40 μmol/L	MGC803	Inhibition of proliferation, invasiveness and migration, induction of apoptosis ↓ phospho-Akt, phospho-Erk, Bcl-2, ↑ Bax, active caspase -3	[91]
Chalcone	0.5–2 μg/mL	AGS	Inhibition of cell viability, adhesion, migration and invasion ↓ MMP9, MMP2, phospho-FAK, phospho-JNK1/2, NF-kB, phospho-IkBα ↑ IkBα	[92]
Licochalcone A	100 μmol/L	BGC-823	Inhibition of proliferation, induction of apoptosis ↓ GSH/GSSG ratio, phospho-PI3K, phospho-Akt, ↑ ROS production, cleaved PARP, phospho-Erk, phospho-JNK, phospho-p38	[93]
Hydroxysafflor yellow A	25–100 μmol/L	HCT116	Inhibition of proliferation and colony formation, invasion and migration, induction od apoptosis ↓ PCNA, Bcl-2, N-cadherin, vimentin, phospho-Akt ↑ Bax, cleaved caspase -3, E-cadherin, PPARγ, PTEN	[94]
3-(3-((E)-3-(4-hydroxy-3-methoxyphenyl)-2-propenoyl)phenyl)-2-methyl-3,4- dihydro-4-quinazolinone	20–40 μmol/L	HCT116	Induction of apoptosis, S and G2/M cell cycle arrest ↓ Bcl-2, cytosolic Bax, mitochondrial cytochrome *c*, phospho-100α, phospho-100γ, phospho-Akt, Akt, ↑ mitochondrial Bax, cytosolic cytochrome *c*, cleaved caspases -3, -9, cleaved PARP, p21, p27, Skp-1, Skp-2, Ub, phospho-mTOR, mTOR, phospho-p70S6K, phospho-STAT3, STAT3	[95]
(E)-1–(1-Hydroxy-4,5,8-trimethoxynaphthalen-2-yl)-3-(quinolin-6-yl)prop-2-en-1-one	1, 2 μmol/L	MKN1	Induction of apoptosis, inhibition of colony formation, migration and invasion	[98]
3-hydroxy-3′,4,4′,5′-tetra-methoxy-chalcone	0.5, 35, 55 μmol/L	HT-29, HCT116	Inhibition of proliferation, G2/M cell cycle arrest, induction of apoptosis, DNA damage ↓ cdc2, ↑ p21, phospho-p53, DR5, cleaved caspases -3, -8, cleaved PARP, phospho-Akt, phospho-Erk, phospho-p38, COX-2, PGE2	[99]
Cardamonin	10–30 μmol/L	AGS	Inhibition of proliferation, colony formation and migration, induction of apoptosis ↓ Bcl-2, Cdk1, cyclin B1, cdc25C, α-SMA, vimentin, Snail, phospho-STAT3, EMT ↑ Bax, caspase -3, p21, E-cadherin,	[100]
1,2,3-triazole linked ciprofloxacin-chalcone	0.1–100 μmol/L	HCT116	Inhibition of proliferation, G2/M cell cycle arrest, inhibition of topoisomerase I and II and tubulin polymerization ↓ α-tubulin ↑ γH2AX, phospho-ATR, phospho-Chk1, phospho-cdc25C	[103]
Xanthohumol	2–8 μmol/L	HCT116, SW6320	Inhibition of colony formation and proliferation ↓ HK1, HK2, phospho-histone H3, cytosolic Bax, phospho-EGFR, phospho-Akt, phospho-Erk1/2 ↑ cleaved caspase -3, cytosolic cytochrome *c*, mitochondrial Bax, mitochondrial VDAC1	[119]
3-phenyl-1-(2,4,6- tris(methoxymethoxy)phenyl)prop-2-yn-1-one)	4 μmol/L	HCT116, SW6320	Inhibition of migration, invasion and proliferation ↓ nuclear NF-kB, phospho-IkBα, MMP-7, ↑ IkBα, HO-1, p21, cleaved PARP, cleaved caspases -3, -8, -9	[120]
Cardamonin	10 μmol/L	HT-29, HCT116	Inhibition of migrtion, invasion and proliferation ↓ MMP-2, MMP-9, N-cadherin, ADRB2, EMT ↑ E-cadherin	[121]
(E)-3-Phenyl-1-(2-Pyrrolyl)-2-Propenone	10–100 μmol/L	HEK293, RAW264.7	↓ iNOS, TNF-α, p50, c-Jun, c-Fos, phospho-IkBα, IkBα, phospho-Syk, phospho-Src, phospho-JNK, phospho-Erk, phospho-IRAK1, phospho-c-Raf, phospho-TAK1, phospho-MEK1, phospho-MKK4	[124]
(Z)-3–(6-bromo-1H-indol-3-yl)-2-fluoro-1–(3,4,5-trimethoxyphenyl)prop-2-en-1-one (4c)	18–72 nmol/L	MGC-803, HUVEC	Inhibition of tubulin polymerization, G2/M phase cell cycle arrest, induction of apoptosis, inhibition of migration and invasion ↓ cyclin B1, phospho-cdc2, MMP ↑ p21, cleaved caspases -3, -7, -9, cleaved PARP, ROS generation	[144]
Isoliquiritigenin	0.4–1.6 μmol/L	RAW264.7	↓ iNOS, COX-2, IL-6, TNF-α, phospho-IkBα, nuclear p65, nuclear p50, phospho-IKK, phospho-Erk, phospho-p38	[161]
Licochalcone A	2.5–20 μmol/L	RAW264.7	↓ iNOS, COX-2, PGE_2_, IL-1β, IL-6, phospho-IkBα, phospho-p38, nuclear NF-kB, ↑ IL-10, IkBα, AP-1 DNA-binding activity	[162]
Bromo chalcone derivative H72	1–6 μmol/L	MGC803, HGC27	Inhibition of proliferation, induction of apoptosis, ROS production ↓ Bid, Bcl-xL, XIAP, survivin, MMP ↑ Bim, DR4, DR5, cytochrome *c*, cleaved PARP, cleaved caspases -3, -9	[183]
Flavokawain A derivate S17	10 μmol/L	MGC803	Inhibition of proliferation, induction of apoptosis, ROS production ↓ p18, XIAP, Nrf2 ↑ cleaved caspases -3, -8, -9, cleaved PARP, p43/p41, Bim, Bax, Bid, Bad, DR5, phospho-Nrf2, HO-1, NQO1	[184]
Flavokawain C	40–80 μmol/L	HCT116, HT-29	Induction of apoptosis, inhibition of proliferation, G2/M phase cell cycle arrest, ROS production ↓ MMP, c-IAP1, c-IAP2, XIAP ↑ activity of caspases -3, -8, -9, cleaved PARP, p21, p27, GADD153	[187]

**Table 2 ijms-24-05964-t002:** In vivo antitumor effect of selected chalcones.

Chalcone	Dose	Exposure Duration	Tumor Model	Antitumor Effect	Reference
Chalcone derivative L2H17	25–50 mg/kg	60 days	CT26.WT tumor induced in mouse model	Extended survival	[23]
(E)-2-Chloro-4′-methoxychalcone,	15 mg/kg	7 days	ICR and C57BL/6 (B6) mice	Inhibition of TNF-α, IL-1β, IL-6 and IL-12 expression and prolonged survival in LPS-induced acute inflammatory model	[194]
Flavokawain B	Intraperitoneal (1.5 mg/kg) and oral (7.5 mg/kg)	51 days	AGS-xenografted nude mice	Inhibition of tumor growth, reduction of tumor volume and weight, prolonged the survival rate and induced autophagy ↑ LCI/II, Beclin-1, ATG7	[53]
2′-dihydroxy-4,4′-dimethoxydihydrochalcone	12-48 mg/kg	32 days	MKN45 xenograftedBALB/C-nu mice	Significant decrease in tumor volumes, inhibition of Ki67 expression	[59]
Liquiritin	15 mg/kg	21 days	SGC7901/DDP xenografted BALB/C-nu mice	Reduced the size, volume and weight of tumor, inhibition of Ki67 expression, induction of apoptosis and autophagy ↓ cyclin D1, Cdk4, p62 ↑ p21, p53, cleaved caspases -3, -8, -9, cleaved PARP, LC3I/II, Beclin-1	[60]
Cardamonin	25 mg/kg	30 days	BGC-823/5-FU xenograft mouse model	Reduced tumor volumes and tumor weights	[73]
Licochalcone A	15, 50 mg/kg	30 days	HCT116 cells in a xenograft model (BALB/c nude mice)	Inhibition of tumor growth, reduced levels of PD-L1, p65, Ras and VEGF in tumor tissues	[75]
Licochalcone A	200, 400 µmol/L	30 days	BGC-823 gastric carcinoma in KM mice	Tumor growth inhibition	[93]
3-(3-((E)-3-(4-hydroxy-3-methoxyphenyl)-2-propenoyl)phenyl)-2-methyl-3,4- dihydro-4-quinazolinone	30, 40 mpk	13 days	Ehrlich Ascites Carcinoma cells in Swiss albino mice/Sarcoma-180 in BALB/c mice	Tumor growth inhibition and reduced tumor volume/weight	[95]
(E)-1–(1-Hydroxy-4,5,8-trimethoxynaphthalen-2-yl)-3-(quinolin-6-yl)prop-2-en-1-one	5 mg/kg	28 days	MKN1 xenografted model (BALA/c nude mice)	Tumor growth inhibition and reduced tumor and metastasis mass	[98]
Xanthohumol	10 mg/kg	16 days	HT29 and HCT116 xenograft mouse model	Reduced tumor volume and weight, decreased phosphorylation of Akt, expression of Ki67 and HK2	[119]
Licochalcone A	1–10 mg/kg	5 days	LPS-treated BALB/c mice	Extended survival, reduced plasmatic concentrations of NO, IL-6, TNF-α	[162]
Bromo chalcone derivative H72	40 mg/kg/day	21 days	MGC803 xenograft model	Reduced tumor weight	[183]
Flavokawain A derivate S17	40 mg/kg/day	24 days	MGC803 xenograft model	Lowered tumor growth rate, increased expression od cleaved caspases -3 and -8, ROS production	[184]
Licorice Extracts Containing Enhanced Licochalcone A	25–100 mg/kg	24 weeks	H. pylori-infected animal model (mice)	Reduced inflammation, mucosal erosion/ulcer, dysplasia, adenoma formation ↓ bFGF, FcrRIIB, ICAM-1, lungkine, thymus-CK1, TRANCE, TROY, COX-2, iNOS, TNF-α, IL-1β, IL-6, phospho-STAT3, phospho-JAK2	[248]
Lotus leaf flavonoids extract	25–100 mg/kg	2 weeks	H. pylori-infected BALB/c mice	Reduction of gastric lesions, decreased levels of endothelin-1, substance P, increased levels of somatostatin, and vasoactive intestinal peptide, IL-6, IL-12, TNF-α, INF-γ, MPO, KRT16, KRT6b, TGM3, NLRP3, IL-1β, TLR4	[249]
Xanthohumol	5 mg/kg	8 weeks	Sprague-Dawley rats	Inhibition of formation of aberrant crypt foci, reduced MDA level, reduced expression of iNOS and COX-2, cyclin D1, c-Myc and Bcl-2, suppression of Wnt/β-catenin signaling and Ki76 positive cells, increased Bax and caspase -3, SOD, CAT, GPx	[235]
Licochalcone A	5–30 mg/kg	8 weeks	Licochalcone A	Reduction of tumor formation and tumor volume, decreased cytokine levels in colon (IL-1β, IL-6, TNF-α, KC, MCP-1), inhibition of IκBα and expression of iNOS, COX-2, β-catenin and MMP-9	[234]

## Data Availability

Not applicable.

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
