# Peer review of "Chalcones and Gastrointestinal Cancers: Experimental Evidence"

_ijms, 2023, doi:10.3390/ijms24065964_

Round 1

Reviewer 1 Report

It is better to add a table containing in vitro and in vivo findings.

Author Response

Please, see an attached file.

Reviewer 2 Report

In this review paper, Michalkova et al discussed the role and mechanism of chalcone and its derivatives in gastrointestinal cancers, which should normally cover both colon and gastric cancers. Well, in many of the sections, they detailed several aspects of colorectal cancer, however, there is only one paragraph about gastric cancer. Based on this, they should either write about gastric cancer in all the other sections, or they should change the title of the article to colorectal cancer alone and remove all parts pertaining to gastric cancer. 

On the other hand, I noted that this review paper is not very well organized, for example, it is necessary to include the role of H. pylori in the gastric section, instead of it being a separate one. Regarding the mechanism, the authors should prepare a figure for each mechanism or at least include them all in two different figures. Vis-à-vis cancer progression and invasion, we know that it occurs via epithelial-mesenchymal transition, therefore, several papers reported the outcome of chalcones and its derivatives on EMT, nevertheless, the authors did not dedicate a section on this important topic. 

The authors should cite these three important papers:

-       Rizeq B, Gupta I, Kheraldine H, Elkhalifa D, Al-Farsi HF, Moustafa AA, Khalil A. Novel Nitrogen-Based Chalcone Analogs Provoke Substantial Apoptosis in HER2-Positive Human Breast Cancer Cells via JNK and ERK1/ERK2 Signaling Pathways. Int J Mol Sci. 2021 Sep 6;22(17):9621. doi: 10.3390/ijms22179621. PMID: 34502529; PMCID: PMC8431802.

-       Elkhalifa D, Al-Hashimi I, Al Moustafa AE, Khalil A. A comprehensive review on the antiviral activities of chalcones. J Drug Target. 2021 Apr;29(4):403-419. doi: 10.1080/1061186X.2020.1853759. Epub 2020 Dec 16. PMID: 33232192.

-       Elkhalifa D, Siddique AB, Qusa M, Cyprian FS, El Sayed K, Alali F, Al Moustafa AE, Khalil A. Design, synthesis, and validation of novel nitrogen-based chalcone analogs against triple-negative breast cancer. Eur J Med Chem. 2020 Feb 1;187:111954. doi: 10.1016/j.ejmech.2019.111954. Epub 2019 Dec 7. PMID: 31838326.

Finally, the paper needs revision for English.

Author Response

Please, see an attached file.

Reviewer 3 Report

1.   In P.1 lines 11-13, “The current treatment including surgery, chemotherapy or radiotherapy has several limitations such as drug toxicity, cancer recurrence or drug resistance and thus it a great challenge to …” should be corrected as “The current treatments including surgery, chemotherapy or radiotherapy have several limitations such as drug toxicity, cancer recurrence or drug resistance and thus it’s a great challenge to…”.

2.  In P.1 line 32 “Nature was, is and it will be an important source …” should be corrected as “Nature was, is and will be an important source …”.

3. In P.1 lines 40-41, “.. beneficial effects of these compounds on human health including anticancer, anti-inflammatory, antioxidant and cardioprotective” should be changed as  “.. beneficial effects of these compounds on human health including anticancer, anti-inflammation, antioxidation and cardioprotection”.

4. In P.2 lines 48-9, “Moreover, the anticancer effect of both natural and synthetic chalcones has also been intensively studied” should be modified as “Moreover, the anticancer effects of both natural and synthetic chalcones have also been intensively studied”.

5. In P.2, line 66, “.. block cancer cell proliferation via cell cycle arrest and activation of cell death machinery” should be modified as “.. block cancer cell proliferation via inducing cell cycle arrest and activating cell death machinery”.  

6. In P.3 lines 78-9, “.. the expression of tubulins was deregulated on both genomic and protein levels” should be corrected as “.. the expression of tubulins was deregulated on both gene and protein levels”.

7. In P.3 lines 125-6, “Similarly, to colon cancer cells, chalcones mostly stop cell cycle progression at the G2/M phase also in gastric cancer cells” should be changed as “Similarly to colon cancer cells, chalcones stop cell cycle progression in gastric cancer cells mostly at the G2/M phase”.  

8. In P.4 lines 143-5 “.. that several chalcones also iniciate cell death via activation of extrinsic pathway of apoptosis [54-57] or induction of non-apoptotic cell death” should be corrected as “.. that several chalcones also initiate cell death via activation of the extrinsic pathway of apoptosis [54-57] or induction of the non-apoptotic cell death”.

9. In P.5 lines 184-6 “via inhibition of Wnt/β-catenin signaling pathway in BGC-823 and BGC-823 5-fluorouracil (5-FU) resistant gastric cancer cells” could be modified as “via inhibition of Wnt/β-catenin signaling pathway in the 5-fluorouracil (5-FU) resistant BGC-823 and BGC-823 gastric cancer cells”.

10. In P.6 lines 249-50 “.. associated with the down-regulation of matrix metalloprotein-ase-2 and -9 expression, enzymes involved in extracellular matrix degradation” should be modified as “.. associated with the down-regulated expression of matrix metalloproteinase-2 and -9, the enzymes involved in extracellular matrix degradation”.

11.  In P.7 lines 268-9 “.. and consequence of its inhibition is cancer cell cycle arrest, cell death, suppression of neovascularisation or modulation of immune response” should be modified as “.. and the consequences of its inhibition include cancer cell cycle arrest, cell death, suppression of neovascularisation or modulation of immune response”.  

12.  In P.7 line 281 “The process of neovascularisation is gently orchestrated by numerous positive and …” could be modified as “The process of neovascularisation is coordinatively orchestrated by numerous positive and …”.

13.  In P.7 line 289 “As it was mentioned above VEGFR-2 is an important target for antiangiogenic drugs” should be shortened as “As aforementioned, VEGFR-2 is an important target for antiangiogenic drugs”.

14.  In P. 8 line 357 “Their antimicrobial, antibacterial and antiparasitic potential has been proven in relation to CRC” should be corrected as “Their antimicrobial and antiparasitic potentials have been proven in relation to anti-CRC effects”.

 15.  In P. 9 lines 410-11 “human origin intestinal epithelial cell (IEC) HT-29 cells” should be corrected as “HT-29 human CRC cells”. 

Author Response

Please, see an attached file.

Round 2

Reviewer 2 Report

The authors did not address my comments correctly.  

Author Response

Dear reviewer,

We again tried to find more publications related to chalcones,” “EMT,” “colorectal cancer “ and „gastric cancer“ , however, without significant success. In an effort to meet your recommendations, we have added a mention of EMT to the relevant works cited in the manuscript.

In addition, we included one of the recommended references in the introduction of the manuscript.